# Sharp Matrix Empirical Bernstein Inequalities

**Hongjian Wang**
Department of Statistics and Data Science
Carnegie Mellon University
Pittsburgh, PA 15213
hjnwang@cmu.edu

**Aaditya Ramdas**
Department of Statistics and Data Science
Carnegie Mellon University
Pittsburgh, PA 15213
aramdas@cmu.edu

## Abstract

We present two sharp, closed-form empirical Bernstein inequalities for symmetric random matrices with bounded eigenvalues. By sharp, we mean that both inequalities adapt to the unknown variance in a tight manner: the deviation captured by the first-order $1/\sqrt{n}$ term asymptotically matches the matrix Bernstein inequality exactly, including constants, the latter requiring knowledge of the variance. Our first inequality holds for the sample mean of independent matrices, and our second inequality holds for a mean estimator under martingale dependence at stopping times.

## 1 Introduction

We are interested in nonasymptotic confidence sets for the common mean of independent or martingale-dependent bounded random matrices that optimally adapt to the unknown underlying variance. We first review the scalar case to set some context.

### 1.1 Background: Scalar Empirical Bernstein Inequalities

The classical Bennett-Bernstein inequality (see Lemma 5 of Audibert et al. [2009]; also Appendix B.1) states that, for the average $\overline{X}_n$ of independent random scalars $X_1, \ldots, X_n$ with common expected value $\mu = \mathbb{E}X_i$, common almost sure upper bound $|X_i| \leqslant B$, and second moment upper bound $\sum_{i=1}^{n} \mathbb{E}X_i^2 \leqslant n\sigma^2$,

$$\mathbb{P}\left(\overline{X}_n - \mu \geqslant \frac{B\log(1/\alpha)}{3n} + \sqrt{\frac{2\sigma^2 \log(1/\alpha)}{n}}\right) \leqslant \alpha. \tag{1}$$

It is clear that (1) remains true if the assumptions are centered instead: $|X_i - \mu| \leqslant B$ and $\sum_{i=1}^{n} \mathbf{Var}(X_i) \leqslant n\sigma^2$. A crucial feature of (1) is that if $\sigma^2 \approx \mathbb{E}X_1^2 \ll B^2$, the deviation is dominated by the "variance term" $\Theta\left(\sqrt{n^{-1}\sigma^2 \log(1/\alpha)}\right)$, tighter than the "boundedness term" $\Theta(\sqrt{n^{-1}B^2 \log(1/\alpha)})$ that dominates if Hoeffding's inequality [1963] is applied instead in the absence of the variance bound $\sigma^2$.

In practice, one often knows $B$ but has no prior access to the possibly much smaller $\sigma$. Thus, such bounds are usually only used in theoretical analysis, but not to practically construct confidence bounds for the mean. For the latter task, so-called nonasymptotic *empirical* Bernstein (EB) inequalities are therefore of particular interest. These inequalities often only assume the almost sure upper bound $B$ of the random variables and are agnostic and *adaptive* to the true variances $\mathbf{Var}(X_i)$, to the effect that the final deviation is still dominated by an asymptotically $\Theta\left(\sqrt{n^{-1}\sigma^2 \log(1/\alpha)}\right)$ variance term, instead of $\Theta(\sqrt{n^{-1}B^2 \log(1/\alpha)})$ from Hoeffding's inequality under the *same* boundedness assumption.

39th Conference on Neural Information Processing Systems (NeurIPS 2025).

Scalar EB inequalities are derived from two very different types of techniques. First, a union bound between a non-empirical ("oracle") Bernstein inequality and a concentration inequality on the sample variance, which is employed by early empirical Bernstein results [Audibert et al., 2009, Maurer and Pontil, 2009]. For example, for i.i.d., $[0,1]$-bounded $X_1, \ldots, X_n$, and their Bessel-corrected sample variance $\hat{\sigma}_n^2$, Maurer and Pontil [2009, Theorem 4] prove the EB inequality

$$\mathbb{P}\left(\overline{X}_n - \mu \geqslant \sqrt{\frac{2\hat{\sigma}_n^2 \log(2/\alpha)}{n}} + \frac{7\log(2/\alpha)}{3(n-1)}\right) \leqslant \alpha. \tag{2}$$

Second, the self-normalization martingale techniques of Howard et al. [2021, Theorem 4] and Waudby-Smith and Ramdas [2023], which enable sharper rates, stopping time-valid concentration, martingale dependence, and variance proxy by predictable estimates other than the sample variance. For example, Waudby-Smith and Ramdas [2023, Theorem 2, Remark 1] prove the following EB inequality for $[0,1]$-bounded random variables $X_1, \ldots, X_n$ with common conditional mean $\mu = \mathbb{E}[X_i|X_1, \ldots, X_{i-1}]$:

$$\mathbb{P}\left(\hat{\mu}_n - \mu \geqslant \sqrt{\frac{2\log(1/\alpha)V_{n,\alpha}}{n}}\right) \leqslant \alpha. \tag{3}$$

Above, $\hat{\mu}_n$ is a particular weighted average of $X_1, \ldots, X_n$, and $V_{n,\alpha} = V(\alpha, X_1, \ldots, X_n)$ a quantity depending on the sample and $\alpha$ that converges to $\sigma^2$ almost surely in the $n \to \infty$ limit, if $X_1, \ldots, X_n$ have a common conditional variance $\sigma^2$. [1]

These exact terms will become clear when we present our matrix result later in Section 4 (taking $d = 1$), but one can already notice the important fact that (3) matches (1) asymptotically without requiring a known variance bound: letting $D_n = \sqrt{2n^{-1}\log(1/\alpha)V_{n,\alpha}}$ be the deviation term of (3),

$$\sqrt{n}D_n \overset{a.s.}{\to} \sqrt{2\log(1/\alpha)\sigma^2}, \tag{4}$$

a limit also attained by the oracle Bennett-Bernstein inequality (1)'s deviation term. A one-sided scalar EB inequality is said to be *sharp* if its deviation term $D_n$ satisfies the oracular limit (4): its first order term, including constants, asymptotically matches the oracle Bernstein inequality which requires knowledge of $\sigma^2$. We see that while (3) is sharp, (2) is not sharp.

Indeed, these two methods are inherently different and as argued convincingly by Howard et al. [2021, Appendix A.8]: the latter's avoidance of the union bound produces a better concentration. Waudby-Smith and Ramdas [2023] were the first ones to prove that their EB inequality is sharp, pointing out that the union bound-based inequalities are not sharp (but only slightly so). We further discuss this issue in Appendix B.2, showing that one can make the Maurer-Pontil inequality (2) sharp by using a smarter union bound, but it still is empirically looser than (3). Other EB inequalities have been proved in the literature in between the above sets of papers, but they are even looser than the original, so we omit them.

## 1.2 Our Contributions: Matrix Empirical Bernstein Inequalities

Exponential concentration inequalities for the sum of independent matrices are in general much harder to obtain. Tropp [2012, Theorem 6.1] proved a series of Bennett-Bernstein inequalities for the average $\overline{\mathbf{X}}_n$ of independent $d \times d$ symmetric matrices $\mathbf{X}_1, \ldots, \mathbf{X}_n$ with common mean $\mathbb{E}\mathbf{X}_i = \mathbf{M}$, common eigenvalue upper bound $\lambda_{\max}(\mathbf{X}_i) \leqslant B$, and $\sum_{i=1}^n \mathbb{E}\mathbf{X}_i^2 \preceq n\mathbf{V}$. For example, the Bennett-type result implies the following ($\|\cdot\|$ being the spectral norm),

$$\mathbb{P}\left(\lambda_{\max}\left(\overline{\mathbf{X}}_n - \mathbf{M}\right) \geqslant \frac{B\log(d/\alpha)}{3n} + \sqrt{\frac{2\log(d/\alpha)\|\mathbf{V}\|}{n}}\right) \leqslant \alpha. \tag{5}$$

The analogy between (1) and (5) is straightforward to notice, including matching constants. See Appendix B.1 for some remarks on these two non-empirical Bernstein results and a proof of (5). We shall explore some of the techniques by Tropp [2012] later when developing our results.

The main contribution of the current paper is two empirical Bernstein inequalities for matrices derived in analogy to the two methods in the scalar case mentioned earlier. First, we generalize the union bound and plug-in techniques by Audibert et al. [2009], Maurer and Pontil [2009] to matrices and obtain:

---

[1] This was originally proved by Waudby-Smith and Ramdas [2023] under i.i.d. assumption. We prove it under martingale dependence in our matrix result.

**Proposition 1.1** (Theorem 3.1 of this paper, shortened). *Let $n$ be even and $\mathbf{X}_1, \ldots, \mathbf{X}_n$ be i.i.d. symmetric matrices with eigenvalues in $[0,1]$ and mean $\mathbf{M}$. Let $\mathbf{V}_n^*$ be the paired sample variance $n^{-1}((\mathbf{X}_1 - \mathbf{X}_2)^2 + (\mathbf{X}_3 - \mathbf{X}_4)^2 + \cdots + (\mathbf{X}_{n-1} - \mathbf{X}_n)^2)$. Then,*

$$\mathbb{P}\left(\lambda_{\max}\left(\overline{\mathbf{X}}_n - \mathbf{M}\right) \geqslant \sqrt{\frac{2\|\mathbf{V}_n^*\| \log \frac{nd}{(n-1)\alpha}}{n}} + \mathcal{O}\left(\frac{\log(nd/\alpha)}{n}\right)\right) \leqslant \alpha. \tag{6}$$

Second, we provide a faithful generalization of (3) to the matrix case which we informally state as follows.

**Proposition 1.2** (Corollary 4.3 of this paper, shortened). *Let $\mathbf{X}_1, \ldots, \mathbf{X}_n$ be symmetric matrices with eigenvalues in $[0,1]$ and common conditional mean $\mathbf{M} = \mathbb{E}[\mathbf{X}_i | \mathbf{X}_1, \ldots, \mathbf{X}_{i-1}]$. For an appropriate weighted average $\hat{\mathbf{M}}_n$ of $\mathbf{X}_1, \ldots, \mathbf{X}_n$ and an appropriate sample variance proxy $v_{n,\alpha} = v(\alpha, \mathbf{X}_1, \ldots, \mathbf{X}_n) > 0$,*

$$\mathbb{P}\left(\lambda_{\max}(\hat{\mathbf{M}}_n - \mathbf{M}) \geqslant \sqrt{\frac{2\log(d/\alpha)v_{n,\alpha}}{n}}\right) \leqslant \alpha. \tag{7}$$

*Further, if $\{\mathbf{X}_i\}$ have a common conditional variance $\mathbf{V}$, $v_{n,\alpha}$ converges almost surely to $\|\mathbf{V}\|$.*

The detailed description of $\hat{\mathbf{M}}_n$ and $v_{n,\alpha}$ will be furnished in Section 4. As in the scalar case, we say a one-sided matrix EB inequality is *sharp* if its deviation term $D_n$ satisfies

$$\sqrt{n}D_n \overset{a.s.}{\to} \sqrt{2\log(d/\alpha)\|\mathbf{V}\|}, \tag{8}$$

as does that of the oracle inequality (5). We see that both results above are sharp matrix empirical Bernstein inequalities. It is also worth remarking that both (3) and our matrix generalization (7) are special fixed-time cases of some *time-uniform* concentration inequalities that control the tails of all $\{\hat{\mu}_n\}_{n \geqslant 1}$ or $\{\hat{\mathbf{M}}_n\}_{n \geqslant 1}$ simultaneously, enabling sequential analysis. This will become clear as we develop our results.

In applications e.g. covariance estimation, our matrix EB inequalities can lead to tremendous improvements over the oracle matrix Bernstein inequality (5). To the best of our knowledge, the only other matrix EB inequality in the literature is the contemporaneous result by Kroshnin and Suvorikova [2024, Corollary 3.5], which is not sharp. Besides the work cited above, some other authors have also contributed to the literature of Bernstein or empirical Bernstein inequalities. These include EB inequalities for vectors by Chugg et al. [2025], for Banach space elements by Martinez-Taboada and Ramdas [2024]; a time-uniform oracle matrix Bernstein inequality by Howard et al. [2021]; a dimension-free version of (5) by Minsker [2017]; and another oracle matrix Bernstein inequality by Mackey et al. [2014]. Some of these will be discussed in Section 5. We also discuss some other closed-form scalar EB inequalities by Tolstikhin and Seldin [2013], Mhammedi et al. [2019], Mhammedi [2021], Jang et al. [2023], Orabona and Jun [2024], Shekhar and Ramdas [2023] in Appendix C.

## 2   Preliminaries

**Notation**   Let $\mathcal{S}_d$ denote the set of all $d \times d$ real-valued symmetric matrices, which is the only class of matrices considered in this paper. These matrices are denoted by bold upper-case letters $\mathbf{A}, \mathbf{B}$, etc. For $I \subseteq \mathbb{R}$, we denote by $\mathcal{S}_d^I$ the set of all real symmetric matrices whose eigenvalues are all in $I$. $\mathcal{S}_d^{[0,\infty)}$, the set of positive semidefinite and $\mathcal{S}_d^{(0,\infty)}$, the set of positive definite matrices are simply denoted by $\mathcal{S}_d^+$ and $\mathcal{S}_d^{++}$ respectively. The Loewner partial order is denoted $\preceq$, where $\mathbf{A} \preceq \mathbf{B}$ means $\mathbf{B} - \mathbf{A}$ is positive semidefinite, and $\mathbf{A} \prec \mathbf{B}$ means $\mathbf{B} - \mathbf{A}$ is positive definite. We use $\lambda_{\max}$ to denote the largest eigenvalue of a matrix in $\mathcal{S}_d$, and $\| \cdot \|$ its spectral norm, i.e., the largest absolute value of eigenvalues. As is standard in matrix analysis, a scalar-to-scalar function $f : I \to J$ is identified canonically with a matrix-to-matrix function $f : \mathcal{S}_d^I \to \mathcal{S}_d^J$, following the definition

$$f : \mathbf{U}^\mathsf{T} \operatorname{diag}[\lambda_1, \ldots, \lambda_d]\mathbf{U} \mapsto \mathbf{U}^\mathsf{T} \operatorname{diag}[f(\lambda_1), \ldots, f(\lambda_d)]\mathbf{U}. \tag{9}$$

Matrix powers $\mathbf{X}^k$, logarithm $\log \mathbf{X}$, and exponential $\exp \mathbf{X}$ are common examples. It is worth noting that the monotonicity of $f : I \to J$ is usually *not* preserved when lifted to $f : \mathcal{S}_d^I \to \mathcal{S}_d^J$ in the $\preceq$

order. The matrix logarithm, however, is monotone. On the other hand, for any monotone $f : I \to J$, the function $\mathrm{tr} \circ f : \mathcal{S}_d^I \to \mathbb{R}$ is always monotone.

We work on a filtered probability space $(\Omega, \mathcal{F}, \mathbb{P})$ where $\mathcal{F} := \{\mathcal{F}_n\}_{n \geqslant 1}$ is a filtration, and we assume $\mathcal{F}_0 := \{\varnothing, \Omega\}$. We say a process $X := \{X_n\}$ is adapted if $X_n$ is $\mathcal{F}_n$-measurable for all integers $n \geqslant 0$ or sometimes $n \geqslant 1$; predictable if $X_n$ is $\mathcal{F}_{n-1}$-measurable for all integers $n \geqslant 1$.

**Nonnegative Supermartingales**   Many of the classical concentration inequalities for both scalars and matrices are derived via Markov's inequality. Howard et al. [2020] pioneered using *Ville's inequality* for *nonngative supermartingales* to construct time-uniform concentration inequalities. An adapted scalar-valued process $\{X_n\}_{n \geqslant 0}$ is called a nonnegative supermartingale if $X_n \geqslant 0$ and $\mathbb{E}[X_{n+1}|\mathcal{F}_n] \leqslant X_n$ for all $n \geqslant 0$ (all such inequalities are intended in the $\mathbb{P}$-almost sure sense). Let us state the following two well-known forms of Ville's inequality, both generalizing Markov's inequality.

**Lemma 2.1** (Ville's inequality). *Let $\{X_n\}$ be a nonnegative supermartingale and $\{Y_n\}$ be an adapted process such that $Y_n \leqslant X_n$ for all $n$. For any $\alpha \in (0,1]$,*

$$\mathbb{P}\left(\sup_{n \geqslant 0} Y_n \geqslant X_0/\alpha\right) \leqslant \alpha. \tag{10}$$

*Equivalently, for any stopping time $\tau$,*

$$\mathbb{P}\left(Y_\tau \geqslant X_0/\alpha\right) \leqslant \alpha. \tag{11}$$

**Matrix CGF Supermartingales**   The Chernoff-Cramér MGF method cannot be directly applied to the sum of independent random matrices due to $\exp(\mathbf{A} + \mathbf{B}) \neq (\exp \mathbf{A})(\exp \mathbf{B})$ in general. Tropp [2012] introduced the method of controlling the *trace* of the matrix CGF via an inequality due to Lieb [1973]. The Lieb-Tropp method is later furthered by Howard et al. [2020] in turn to construct a nonnegative supermartingale for matrix martingale differences. We slightly generalize it as follows.

**Lemma 2.2** (Lemma 4 in Howard et al. [2020], rephrased and generalized). *Let $\{\mathbf{Z}_n\}$ be an $\mathcal{S}_d$-valued, adapted martingale difference sequence. Let $\{\mathbf{C}_n\}$ be an $\mathcal{S}_d$-valued adapted process, $\{\mathbf{C}'_n\}$ be an $\mathcal{S}_d$-valued predictable process. If*

$$\mathbb{E}(\exp(\mathbf{Z}_n - \mathbf{C}_n)|\mathcal{F}_{n-1}) \preceq \exp(\mathbf{C}'_n), \tag{12}$$

*holds for all $n$, then the process*

$$L_n = \mathrm{tr} \exp\left(\sum_{i=1}^n \mathbf{Z}_i - \sum_{i=1}^n (\mathbf{C}_i + \mathbf{C}'_i)\right) \tag{13}$$

*is a nonnegative supermartingale. Further,*

$$L_n \geqslant \exp\left(\lambda_{\max}\left(\sum_{i=1}^n \mathbf{Z}_i\right) - \lambda_{\max}\left(\sum_{i=1}^n (\mathbf{C}_i + \mathbf{C}'_i)\right)\right). \tag{14}$$

We remark that in the supermartingale (13), since the empty sum is the zero matrix, $L_0 = \mathrm{tr} \exp 0 = \mathrm{tr} \mathbf{I} = d$. This will translate into the $\log(d)$-type dimension dependence in our bounds. The above lemma is proved in Appendix A.3.

## 3   First Matrix EB Inequality: Plug-In

The scalar EB inequality (2) by Maurer and Pontil [2009, Theorem 4] is derived via a union bound between the non-empirical Bennett-Bernstein inequality (1) and a lower tail bound on the Bessel-corrected sample standard deviation. We slightly deviate from their construction by restricting the sample size $n$ to even numbers (discarding an observation if $n$ is odd) and considering the following "paired" variance estimator

$$\mathbf{V}_n^* = \frac{1}{n}((\mathbf{X}_1 - \mathbf{X}_2)^2 + (\mathbf{X}_3 - \mathbf{X}_4)^2 + \cdots + (\mathbf{X}_{n-1} - \mathbf{X}_n)^2). \tag{15}$$

Our first matrix EB inequality follows from applying the non-empirical matrix Bennett-Bernstein inequality *twice*, to the sample average and the paired variance estimator above.

**Theorem 3.1** (First matrix empirical Bernstein inequality). *Let $n \geqslant 2$ be even and $\mathbf{X}_1, \ldots, \mathbf{X}_n$ be $\mathcal{S}_d^{[0,1]}$-valued independent random matrices with common mean $\mathbf{M}$ and variance $\mathbf{V}$. We denote by*

$\overline{\mathbf{X}}_n$ their sample average and by $\mathbf{V}_n^*$ their paired variance estimator defined in (15). Then, for any $\alpha \in (0, 1)$,

$$\mathbb{P}\left(\lambda_{\max}\left(\overline{\mathbf{X}}_n - \mathbf{M}\right) \geqslant D_n^{\mathsf{meb1}}\right) \leqslant \alpha, \tag{16}$$

where

$$D_n^{\mathsf{meb1}} = \frac{\log \frac{nd}{(n-1)\alpha}}{3n} + \sqrt{\frac{2\|\mathbf{V}_n^*\| \log \frac{nd}{(n-1)\alpha}}{n}} + \left(\sqrt{\frac{5}{3}} + 1\right) \frac{\sqrt{\left(\log \frac{nd}{(n-1)\alpha}\right)\left(\log \frac{2nd}{\alpha}\right)}}{n}. \tag{17}$$

Further, if $\mathbf{X}_1, \dots, \mathbf{X}_n$ are i.i.d.,

$$\lim_{n\to\infty} \sqrt{n}D_n^{\mathsf{meb1}} = \sqrt{2\log(d/\alpha)\|\mathbf{V}\|}, \quad \textit{almost surely}. \tag{18}$$

*Proof Sketch.* The inequality follows from the following lower tail bound on $\|\mathbf{V}_n^*\|^{1/2}$

$$\mathbb{P}\left(\|\mathbf{V}\|^{1/2} \leqslant \|\mathbf{V}_n^*\|^{1/2} + \left(\sqrt{\frac{5}{6}} + \frac{1}{\sqrt{2}}\right)\sqrt{\frac{\log(2d/\alpha)}{n}}\right) \geqslant 1 - \alpha, \tag{19}$$

and the matrix Bennett-Bernstein inequality (5) via an $\alpha = \alpha(n-1)/n + \alpha/n$ union bound. The full proof can be found in Appendix A.4. $\qquad\square$

The first order term of the deviation radius (17) matches the oracle matrix Bernstein inequality (5), both being the $\Theta\left(\sqrt{n^{-1}\|\mathbf{V}\|\log(d/\alpha)}\right)$ variance term. More importantly, the match is precise asymptotically, as is indicated by the limit (18) of $\sqrt{n}D_n^{\mathsf{meb1}}$. It is therefore a sharp matrix EB inequality by our standard. Indeed, this owes much to the imbalanced $\alpha = \alpha(n-1)/n + \alpha/n$ split in the union bound in the proof; if a balanced, or more generally $n$-independent split was employed, the limit would become $\sqrt{2\log(Cd/\alpha)\|\mathbf{V}\|}$ for some constant $C > 1$ instead. A balanced split, however, is exactly what Maurer and Pontil [2009] do in their scalar EB inequality (as well as Kroshnin and Suvorikova [2024] concurrently in their matrix EB inequality), leading to the intralogarithmic factor $C = 2$ as shown in (2). This non-sharpness of the scalar Maurer-Pontil inequality (2), we remark, can be avoided as well by switching to the $\alpha = \alpha(n-1)/n + \alpha/n$ imbalanced split instead, which leads to both theoretical sharpness and a significant boost in its large-sample empirical performance. We write it down formally and perform comparative simulations in Appendix B.2.

Beyond sharpness, the second order term of the deviation radius (17) also has the same rate as the oracle matrix Berntein inequality (5) up to a logarithmic factor in $n$, both being the boundedness term that decays as $\tilde{\Theta}(n^{-1})$ as the sample size $n$ grows. The second order term of (17) also matches the second order term of the *sharpened* Maurer-Pontil inequality derived in Appendix B.2, with only a slight inflation of the constant. We remark that Maurer and Pontil [2009] employ a self-bounding technique on the "classical" Bessel-corrected scalar sample variance $\hat{\sigma}_n^2$, and we are not aware such a technique exists for random matrices, leading us to opt for the paired sample variance (15). In Appendix B.3, we derive an alternative Maurer-Pontil-style matrix EB inequality using the classical matrix Bessel-corrected sample variance and bound it via the matrix Efron-Stein technique due to Paulin et al. [2016]. The resulting matrix EB inequality is still sharp due to the similar first order term, but its second order term inflates from $\tilde{\Theta}(n^{-1})$ to $\tilde{\Theta}(n^{-3/4})$ and has a slightly worse empirical performance.

## 4 Second Matrix EB Inequality: The Supermartingale Method

Our second matrix EB inequality avoids the analysis of the sample variance via the exponential supermartingale technique and opens up for dependent matrices. Let us, following Howard et al. [2020, 2021], Waudby-Smith and Ramdas [2023], define the function $\psi_{\mathrm{E}} : [0, 1) \to [0, \infty)$ as $\psi_{\mathrm{E}}(\gamma) = -\log(1 - \gamma) - \gamma$. The symbol $\psi_{\mathrm{E}}$ is from the fact that it is the cumulant generating function (CGF) of a centered standard exponential distribution. The following lemma is a matrix generalization of Howard et al. [2021, Appendix A.8], which we prove in Appendix A.5

**Lemma 4.1.** *Let $\{\mathbf{X}_n\}$ be an adapted sequence of $\mathcal{S}_d$-valued random matrices with conditional means $\mathbb{E}(\mathbf{X}_n|\mathcal{F}_{n-1}) = \mathbf{M}_n$. Further, suppose there is a predictable and integrable sequence of $\mathcal{S}_d$-valued random matrices $\{\widehat{\mathbf{X}}_n\}$ such that $\lambda_{\min}(\mathbf{X}_n - \widehat{\mathbf{X}}_n) \geqslant -1$. Let*

$$\mathbf{E}_n = \exp(\gamma_n(\mathbf{X}_n - \widehat{\mathbf{X}}_n) - \psi_{\mathrm{E}}(\gamma_n)(\mathbf{X}_n - \widehat{\mathbf{X}}_n)^2), \quad \mathbf{F}_n = \exp(\gamma_n(\mathbf{M}_n - \widehat{\mathbf{X}}_n)), \quad (20)$$

*where $\{\gamma_n\}$ are predictable $(0,1)$-valued scalars. Then,*

$$\mathbb{E}(\mathbf{E}_n|\mathcal{F}_{n-1}) \preceq \mathbf{F}_n. \quad (21)$$

We are now ready to state in full our matrix empirical Bernstein inequality based on the self-normalization technique. The following theorem is stated as a combination of three tools: a non-negative supermartingale, a time-uniform concentration inequality, and an equivalent concentration inequality at a stopping time.

**Theorem 4.2** (Time-uniform and stopped matrix empirical Bernstein inequalities). *Let $\{\mathbf{X}_n\}$ be an adapted sequence of $\mathcal{S}_d$-valued random matrices with conditional means $\mathbb{E}(\mathbf{X}_n|\mathcal{F}_{n-1}) = \mathbf{M}_n$. Let $\{\widehat{\mathbf{X}}_n\}$ be a sequence of predictable and integrable $\mathcal{S}_d$-valued random matrices such that $\lambda_{\min}(\mathbf{X}_n - \widehat{\mathbf{X}}_n) \geqslant -1$ almost surely. Then, for any predictable $(0,1)$-valued sequence $\{\gamma_n\}$,*

$$L_n^{\mathsf{meb2}} = \operatorname{tr}\exp\left(\sum_{i=1}^n \gamma_i(\mathbf{X}_i - \mathbf{M}_i) - \sum_{i=1}^n \psi_{\mathrm{E}}(\gamma_i)(\mathbf{X}_i - \widehat{\mathbf{X}}_i)^2\right) \quad (22)$$

*is a supermartingale. Denote by $\overline{\mathbf{X}}_n^\gamma$ the weighted average $\frac{\gamma_1\mathbf{X}_1 + \cdots + \gamma_n\mathbf{X}_n}{\gamma_1 + \cdots + \gamma_n}$ w.r.t. the positive weight sequence $\{\gamma_n\}$. Then, for any $\alpha \in (0,1)$,*

$$\mathbb{P}\left(\text{there exists } n \geqslant 1, \ \lambda_{\max}\left(\overline{\mathbf{X}}_n^\gamma - \overline{\mathbf{M}}_n^\gamma\right) \geqslant \frac{\log(d/\alpha) + \lambda_{\max}\left(\sum_{i=1}^n \psi_{\mathrm{E}}(\gamma_i)(\mathbf{X}_i - \widehat{\mathbf{X}}_i)^2\right)}{\gamma_1 + \cdots + \gamma_n}\right) \leqslant \alpha; \quad (23)$$

*and for any stopping time $\tau$, $\alpha \in (0,1)$,*

$$\mathbb{P}\left(\lambda_{\max}\left(\overline{\mathbf{X}}_\tau^\gamma - \overline{\mathbf{M}}_\tau^\gamma\right) \geqslant \frac{\log(d/\alpha) + \lambda_{\max}\left(\sum_{i=1}^\tau \psi_{\mathrm{E}}(\gamma_i)(\mathbf{X}_i - \widehat{\mathbf{X}}_i)^2\right)}{\gamma_1 + \cdots + \gamma_\tau}\right) \leqslant \alpha. \quad (24)$$

*Proof.* Due to Lemma 4.1, we can apply Lemma 2.2 with $\mathbf{Z}_n = \gamma_n(\mathbf{X}_n - \mathbf{M}_n)$, $\mathbf{C}_n = \gamma_n(\widehat{\mathbf{X}}_n - \mathbf{M}_n) + \psi_{\mathrm{E}}(\gamma_n)(\mathbf{X}_n - \widehat{\mathbf{X}}_n)^2$, and $\mathbf{C}_n' = \gamma_n(\mathbf{M}_n - \widehat{\mathbf{X}}_n)$ to see that

$$L_n^{\mathsf{meb2}} = \operatorname{tr}\exp\left(\sum_{i=1}^n \gamma_i(\mathbf{X}_i - \mathbf{M}_i) - \sum_{i=1}^n \psi_{\mathrm{E}}(\gamma_i)(\mathbf{X}_i - \widehat{\mathbf{X}}_i)^2\right) \quad (25)$$

is a supermartingale, which upper bounds

$$\exp\left\{\lambda_{\max}\left(\sum_{i=1}^n \gamma_i(\mathbf{X}_i - \mathbf{M}_i)\right) - \lambda_{\max}\left(\sum_{i=1}^n \psi_{\mathrm{E}}(\gamma_i)(\mathbf{X}_i - \widehat{\mathbf{X}}_i)^2\right)\right\}. \quad (26)$$

Applying Lemma 2.1 to (26), the desired result follows from rearranging. $\square$

Before we remark on the uncompromised Theorem 4.2, let us write down its fixed-time, fine-tuned special case of (24) with $\tau = n$ and conditionally homoscedastic observations. This shall justify the "empirical Bernstein" name it bears.

**Corollary 4.3** (Second matrix empirical Bernstein inequality). *Suppose $\alpha \in (0,1)$. Let $\mathbf{X}_1, \ldots, \mathbf{X}_n$ be adapted $\mathcal{S}_d^{[0,1]}$-valued random matrices with constant conditional mean $\mathbf{M} = \mathbb{E}(\mathbf{X}_i|\mathcal{F}_{i-1})$ and constant conditional variance $\mathbf{V} = \mathbb{E}((\mathbf{X}_i - \mathbf{M})^2|\mathcal{F}_{i-1})$. Let $\overline{\mathbf{X}}_i = \frac{1}{i}(\mathbf{X}_1 + \cdots + \mathbf{X}_i)$ and $\overline{\mathbf{X}}_0 = 0$. Define the following variance proxies*

$$\overline{\mathbf{V}}_0 = \frac{1}{4}\mathbf{I}, \quad \overline{\mathbf{V}}_k = \frac{1}{k}\sum_{i=1}^k (\mathbf{X}_i - \overline{\mathbf{X}}_k)^2, \quad \overline{v}_k = \|\overline{\mathbf{V}}_k\| \vee \frac{5\log(d/\alpha)}{n}, \quad (27)$$

*and set $\gamma_i = \sqrt{\frac{2\log(d/\alpha)}{n\overline{v}_{i-1}}}$ for $i = 1, \ldots, n$. Then,*

$$\mathbb{P}\left(\lambda_{\max}\left(\overline{\mathbf{X}}_n^\gamma - \mathbf{M}\right) \geqslant D_n^{\mathsf{meb2}}\right) \leqslant \alpha, \text{ where } D_n^{\mathsf{meb2}} = \frac{\log(d/\alpha) + \lambda_{\max}\left(\sum_{i=1}^n \psi_{\mathrm{E}}(\gamma_i)(\mathbf{X}_i - \overline{\mathbf{X}}_{i-1})^2\right)}{\gamma_1 + \cdots + \gamma_n}.$$

$$(28)$$

*Further, asymptotically,*

$$\lim_{n\to\infty} \sqrt{n} D_n^{\mathsf{meb2}} = \sqrt{2\log(d/\alpha)}\|\mathbf{V}\| \quad \text{almost surely.} \tag{29}$$

We prove Corollary 4.3 in Appendix A.6. The asymptotic behavior (29) of deviation bound $D_n^{\mathsf{meb2}}$ is satisfying as it *adapts fully* to, without knowing, the true variance $\mathbf{V}$. In particular, if the assumption on the known spectral bound is $\mathbf{X}_1, \ldots, \mathbf{X}_n \in \mathcal{S}_d^{[a,b]}$ as opposed to the $\mathcal{S}_d^{[0,1]}$ stated in Corollary 4.3, one can apply the result to $\frac{\mathbf{X}_1 - a}{b-a}, \ldots, \frac{\mathbf{X}_n - a}{b-a}$ to obtain the same

$$\Theta\left(\sqrt{\frac{\log(d/\alpha)\|\mathbf{V}\|}{n}}\right) \tag{30}$$

asymptotic deviation which is free of $b - a$.

The three kinds of result stated in Theorem 4.2 are for potentially different purposes. The super-martingale (22) is best as a sequential test for the null

$$H_0: \ \mathbb{E}(\mathbf{X}_n|\mathcal{F}_{n-1}) = \mathbf{M}_{\mathsf{null}} \quad \text{for all } n \tag{31}$$

by setting each $\mathbf{M}_i$ to $\mathbf{M}_{\mathsf{null}}$. The time-uniform concentration inequality (23) can be used to construct a "confidence sequence" on the common conditional mean $\mathbf{M} = \mathbb{E}(\mathbf{X}_n|\mathcal{F}_{n-1})$; that is, a sequence of confidence balls $B_n = \{\mathbf{M}' \in \mathcal{S}_d : \|\overline{\mathbf{X}}_n^\gamma - \mathbf{M}'\| \leqslant \rho_n\}$ such that $\mathbb{P}(\mathbf{M} \in \cap_n B_n) \geqslant 1 - \alpha$, leading to the stopped concentration inequality (24) which is a valid confidence ball at a fixed stopping time $B_\tau$. We also remark that it is possible to sharpen the confidence ball $B_\tau$ at a fixed stopping time by an *a priori* randomization, due to a recent result by Ramdas and Manole [2024, Theorem 4.1] called "uniformly randomized Ville's inequality". That is, letting $U \sim \mathrm{Unif}_{(0,1)}$ independent from the filtration $\mathcal{F}$, one may replace the $\log(d/\alpha)$ term in (24) with the strictly smaller $\log(Ud/\alpha)$.

The $\mathcal{F}_{i-1}$-measurable term $\widehat{\mathbf{X}}_i$ in Theorem 4.2 is best understood as a "plug-in prediction" of the next observation $\mathbf{X}_i$. Indeed, whereas the inequality holds under all choices of $\widehat{\mathbf{X}}_i$, the smaller the "prediction error" $(\widehat{\mathbf{X}}_i - \mathbf{X}_i)^2$, the tighter the bound. Thus one may set $\widehat{\mathbf{X}}_i$ to be the sample average from $\mathbf{X}_1$ to $\mathbf{X}_{i-1}$, which is exactly what is done in Corollary 4.3.

On the other hand, if the sample size $n$ is *not* fixed in advance and an infinite sequence of i.i.d. (or homoscedastic more generally) observations $\mathbf{X}_1, \mathbf{X}_2, \ldots$, to construct a tight time-uniform concentration bound or powerful sequential test, we recommend setting the weight sequence $\{\gamma_n\}$ as follows: for each sample size $n$, temporarily assume that a sample size of $n$ is fixed in advance, compute the optimal choice of weight on $\mathbf{X}_n$ under this fixed sample size, and set $\gamma_n$ to this optimal choice of weight. This will lead to a vanishing sequence of $\gamma_n = \sqrt{\frac{2\log(d/\alpha)}{n\overline{v}_{n-1}}}$. Under this weight sequence, we see the choice of a *weighted average*

$$\widehat{\mathbf{X}}_n = \overline{\mathbf{X}}_{n-1}^{\psi_{\mathrm{E}}(\gamma)} = \frac{\sum_{i=1}^{n-1} \psi_{\mathrm{E}}(\gamma_i)\mathbf{X}_i}{\sum_{i=1}^{n-1} \psi_{\mathrm{E}}(\gamma_i)} \tag{32}$$

is more reasonable as the weighted sum of squares $\sum_{i=1}^n \psi_{\mathrm{E}}(\gamma_i)(\mathbf{X}_i - x)^2$ is minimized by the weighted average

$$\hat{x} = \frac{\sum_{i=1}^n \psi_{\mathrm{E}}(\gamma_i)\mathbf{X}_i}{\sum_{i=1}^{n-1} \psi_{\mathrm{E}}(\gamma_i)}. \tag{33}$$

Of course, as long as $\widehat{\mathbf{X}}_n$ is any convex combination of $\mathbf{X}_1, \ldots, \mathbf{X}_{n-1}$, the condition $\lambda_{\min}(\mathbf{X}_n - \widehat{\mathbf{X}}_n) \geqslant -1$ is met when $\{\mathbf{X}_n\}$ all take values in $\mathcal{S}_d^{[0,1]}$.

Finally, as a reprise of the shortened version Proposition 1.2 stated in the opening, the "approprioate variance proxy" $v_{n,\alpha} = v(\alpha, \mathbf{X}_1, \ldots, \mathbf{X}_n)$ is simply

$$v_{n,\alpha} = \left(\frac{\log(d/\alpha) + \lambda_{\max}\left(\sum_{i=1}^n \psi_{\mathrm{E}}(\gamma_i)(\mathbf{X}_i - \overline{\mathbf{X}}_{i-1})^2\right)}{\gamma_1 + \cdots + \gamma_n}\right)^2 \frac{n}{2\log(d/\alpha)} \tag{34}$$

which converges almost surely to $\|\mathbf{V}\|$ under conditional homoscedasticity due to (29).

# 5 Comparisons

## 5.1 Self-Normalized EB Inequalities for Scalars and Vectors

Our Theorem 4.2 and Corollary 4.3 owe much to the techniques developed by Waudby-Smith and Ramdas [2023, Theorem 2 and Remark 1] in the scalar case (who in turn build on the earlier result by Howard et al. [2021, Theorem 4] via the "predictable mixing" sequence $\{\gamma_n\}$). In particular, when $d = 1$, our statements match (including constants) exactly the scalar empirical Bernstein inequality counterparts by Waudby-Smith and Ramdas [2023]: Our supermartingale (22) coincides with Equation (13) in Waudby-Smith and Ramdas [2023]; our time-uniform concentration bound (23) becomes identical to Theorem 2 in Waudby-Smith and Ramdas [2023]; and our fixed-time asymptotics (29) recovers Equation (17) in Waudby-Smith and Ramdas [2023]. We also note that Waudby-Smith and Ramdas [2023] assume i.i.d.ness to obtain the fixed-time asymptotics, which, according to our result, can be relaxed to martingale dependence.

As can be expected, applying a vector bound to matrices (by flattening) will lead to a very suboptimal result. The self-normalized empirical Bernstein inequality for vectors due to Chugg et al. [2025, Corollary 5] implies the following for matrices whose Frobenius norm is bounded by $1/2$, for all $\alpha \leqslant 0.1$,

$$\mathbb{P}\left(\|\hat{\mathbf{M}}_n - \mathbf{M}\|_\mathrm{F} \geqslant 3.25\sqrt{\frac{\log(1/\alpha)\tilde{\sigma}_n^2}{n}}\right) \leqslant \alpha. \tag{35}$$

Here, $\tilde{\sigma}_n^2$ converges almost surely to the vectorized variance $\mathbb{E}\|\mathbf{X}_1 - \mathbb{E}\mathbf{M}\|_\mathrm{F}^2$ with i.i.d. matrices. Since everything (assumption and result) is in the Frobenius norm, however, translating the result into the spectral norm will incur a dimensional dependence polynomial in $d$.

Finally, we note that the self-normalized empirical Bernstein inequality for Banach spaces due to Martinez-Taboada and Ramdas [2024] is not applicable as $\mathcal{S}_d$ equipped with the spectral norm is not a 2-smooth Banach space.

## 5.2 Non-Empirical Matrix Bernstein and Hoeffding Inequalities

As we state in the opening (5) and elaborate further in Appendix B.1, Tropp [2012, Theorem 1.4] proves the following matrix Bennett-Bernstein inequality under the assumptions $\max_{1 \leqslant i \leqslant n} \lambda_{\max}(\mathbf{X}_i) \leqslant 1$ and $\sum_{i=1}^n \mathbb{E}\mathbf{X}_i^2 \preceq n\mathbf{V}$:

$$\mathbb{P}\left(\lambda_{\max}\left(\overline{\mathbf{X}}_n - \mathbb{E}\overline{\mathbf{X}}_n\right) \geqslant D_n^\mathsf{tb}\right) \leqslant \alpha, \quad D_n^\mathsf{tb} = \frac{B\log(d/\alpha)}{3n} + \sqrt{\frac{2\log(d/\alpha)\|\mathbf{V}\|}{n}}. \tag{36}$$

We can see that with i.i.d. matrices with variance $\mathbf{V}$, $\sqrt{n}D_n^\mathsf{tb}$ converges to $\sqrt{2\log(d/\alpha)\|\mathbf{V}\|}$ which is the same limit that both $\sqrt{n}D_n^\mathsf{meb1}$ and $\sqrt{n}D_n^\mathsf{meb2}$ converge to, stated as (18) and (29). Therefore, our empirical Bernstein inequalities provide a confidence region fully adaptive to the unknown variance $\mathbf{V}$ and match in asymptotics this oracle Bernstein result which requires $\mathbf{V}$ to be known. Both are thus *sharp* EB inequalities. Assumption-wise, it is important to note that it is fair to compare our $\mathbf{X}_i \in \mathcal{S}_d^{[0,1]}$ assumption to their $\lambda_{\max}(\mathbf{X}_i) \leqslant 1$ assumption; no constant is glossed over in making this comparison when and two-sided bound is sought. To see this, the bound by Tropp [2012, Theorem 1.4] can be applied to $\mathbf{X}_1 - \mathbf{M}, \ldots, \mathbf{X}_n - \mathbf{M}$, and it takes $\mathbf{X}_1 \in \mathcal{S}_d^{[0,1]}$ to ensure both $\lambda_{\max}(\mathbf{X}_1 - \mathbf{M}) \leqslant 1$ and $\lambda_{\max}(-\mathbf{X}_1 + \mathbf{M}) \leqslant 1$ hold.

Mackey et al. [2014, Corollary 5.2] also obtain a matrix Bernstein inequality. However, as they acknowledge in the paper, their bound is strictly looser than the bound by Tropp [2012, Theorem 1.4]. The bound by Minsker [2017, Theorem 3.1] under the same assumption reads

$$\mathbb{P}\left(\lambda_{\max}\left(\overline{\mathbf{X}}_n - \mathbb{E}\overline{\mathbf{X}}_n\right) \geqslant D_n^\mathsf{mb}\right) \leqslant \alpha, \quad D_n^\mathsf{mb} = \frac{B\log(d'/\alpha) + \sqrt{B^2\log^2(d'/\alpha) + 18n\log(d'/\alpha)\|\mathbf{V}\|}}{3n}, \tag{37}$$

where $d' = 14\mathrm{tr}(\mathbf{V})/\|\mathbf{V}\|$, which decides the dimension-free virtue of their result. Matrix Bernstein inequalities that are either anytime-valid or empirical remain an open problem.

Finally, we quote the tightest known Hoeffding-type inequalities for matrices in the literature. Mackey et al. [2014, Corollary 4.2] shows that if independent $\mathbf{X}_1, \ldots, \mathbf{X}_n$ satisfy $(\mathbf{X}_i - \mathbb{E}\mathbf{X}_i)^2 \preceq \mathbf{B}$ almost surely, then

$$\mathbb{P}\left(\lambda_{\max}(\overline{\mathbf{X}}_n - \mathbb{E}\overline{\mathbf{X}}_n) \geqslant \sqrt{\frac{2\|\mathbf{B}\|\log(d/\alpha)}{n}}\right) \leqslant \alpha. \tag{38}$$

A time-uniform extension can be achieved by applying Lemma 3(h) in Howard et al. [2020], but its fixed-time corollary remains identical as (38). The squared boundedness assumption $(\mathbf{X}_i - \mathbb{E}\mathbf{X}_i)^2 \preceq \mathbf{B}$ implies $\mathbf{X}_i - \mathbb{E}\mathbf{X}_i \in \mathcal{S}_d^{[-\|\mathbf{B}\|^{1/2}, \|\mathbf{B}\|^{1/2}]}$, so it is a stronger assumption than the boundedness assumption we make in Corollary 4.3. Further, since $(\mathbf{X}_i - \mathbb{E}\mathbf{X}_i)^2 \preceq \mathbf{B}$ implies $\mathbf{Var}(\mathbf{X}_i) \preceq \mathbf{B}$ and in practice this gap can be arbitrarily large, we see that our empirical Bernstein inequality is asymptotically tighter and the *worst* that can happen is a degradation to this, already tightest, matrix Hoeffding bound, when $\|\mathbf{Var}(\mathbf{X}_i)\| \approx \|\mathbf{B}\|$.

The advantage of matrix EB inequalities becomes even clearer when we consider the recurring application example in the matrix concentration literature: covariance matrix estimation.

**Example 5.1** (Adaptive covariance estimation). Let $X_1, \ldots, X_n$ be i.i.d. random vectors in $\mathbb{R}^d$ with $\|X_1\| \leqslant 1$ almost surely, mean $\mathbb{E}X_1 = 0$ and covariance matrix $\mathbb{E}(X_1 X_1^\mathsf{T}) = \Sigma$. Since

$$\lambda_{\max}(X_1 X_1^\mathsf{T}) = \mathrm{tr}(X_1 X_1^\mathsf{T}) = \mathrm{tr}(X_1^\mathsf{T} X_1) \leqslant 1, \tag{39}$$

we can invoke either of the two matrix EB inequalities (Theorem 3.1 or Corollary 4.3) with $\mathbf{X}_i = X_i X_i^\mathsf{T}$ and $\mathbf{M} = \Sigma$ to construct confidence sets for $\Sigma$ that are adaptive to the unknown 4th moment $\mathbb{E}(\|X_1\|^2 X_1 X_1^\mathsf{T})$.

In comparison, covariance estimation bounds derived via (5) or its variants always have the unknown $\|\Sigma\|$ term in the radius of the concentration, from bounding the 4th moment $\mathbb{E}(\|X_1\|^2 X_1 X_1^\mathsf{T}) \preceq \mathbb{E}(X_1 X_1^\mathsf{T}) = \Sigma$ [Tropp, 2015, Howard et al., 2021]. To turn them into nonasymptotic confidence sets, a further boundedness argument $\|\Sigma\| \leqslant 1$ is required, essentially reducing the Bernstein bound to a Hoeffding bound which can be arbitrarily loose. The sharpness of our EB-based methods immediately benefits the downstream applications of covariance estimation in e.g. machine learning and signal processing.

## 5.3 Simulation

We compare the terms, $D_n^{\mathsf{meb1}}$ of the first matrix empirical Bernstein inequality as in (17), and $D_n^{\mathsf{meb2}}$ of the second matrix empirical Bernstein inequality as in (28), divided by that of the oracle matrix Bennett-Bernstein inequality $D_n^{\mathsf{tb}}$ as in (36). We set $\alpha$ to .05, thus comparing the tightness of one-sided 95%-confidence sets (or equivalently, the spectral diameters of two-sided 90%-confidence sets). The i.i.d. random matrices are generated from 3 fixed orthonormal projections with $d = 3$, each with an independent $\mathrm{Unif}_{[0,1]}$ eigenvalue. The comparison is displayed in Table 1. We see that while for both matrix EB inequalities, the deviations relative to the oracle Bernstein are proved to converge to 1 as $n$ increases, our second matrix EB inequality achieves such "oracular convergence" at a much faster rate. A comparison in the scalar case can be found in Appendix B.2 which conveys a similar message.

| Sample Size $(n)$ | $D_n^{\mathsf{meb1}}/D_n^{\mathsf{tb}}$ | $D_n^{\mathsf{meb2}}/D_n^{\mathsf{tb}}$ |
|---|---|---|
| 100 | 2.612 | 1.397 |
| 1,000 | 1.589 | 1.105 |
| 10,000 | 1.214 | 1.022 |
| 100,000 | 1.072 | 1.007 |
| 1,000,000 | 1.024 | 1.002 |

Table 1: Relative lengths of 95% one-sided confidence sets by two sharp matrix empirical Bernstein inequalities compared to the (oracle) matrix Bennett-Bernstein inequality (5).

## 6 Summary

We provide two new matrix concentration inequalities in this paper. The first one is based on the union bound method, and characterizes, in terms of the paired sample variance, the concentration of the sample mean of independent symmetric matrices with bounded largest eigenvalues, common mean, and common variance. The second one is a self-normalized, time-uniform concentration inequality for the weighted sum of martingale difference symmetric matrices with bounded largest eigenvalues, which when weighted properly, becomes an empirical Bernstein inequality that echoes many of the previous self-normalized-type empirical Bernstein inequalities for scalar, vectors, and Banach space elements.

These two matrix EB inequalities have different advantages: the first one is conceptually simpler, requires only the sample mean and a sample variance, and is computationally easier (both needing $\mathcal{O}(n)$ steps but the first one having smaller constants); the second EB is empirically tighter across all sample sizes, allows martingale dependence, and has a time-uniform version. According to our simulation, the second matrix EB inequality's confidence set is only 10.5% larger compared to the oracle Bernstein inequality under a sample size of 1,000; and 2.2% under a sample size of 10,000. On the other hand, our two matrix EB inequalities both have a closed-form expression, and they match in asymptotics the best non-empirical matrix Bernstein inequality in the literature, as they only depend (in the large sample limit) on the true variance of the matrices which is not required to be known in our bounds, but required in non-empirical bounds. We expect future work to address the challenging problem of unifying our methods with those of the dimension-free matrix Bernstein inequality by Minsker [2017].

## Acknowledgments and Disclosure of Funding

We thank Diego Martinez-Taboada and Arun Kumar Kuchibhotla for helpful discussions. We thank an anonymous referee for pointing out an improvement for Section 3.

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

# A  Additional Proofs

## A.1  Technical Lemmas

The following lemma converts bounds on $|a - b|$ to $\sqrt{a} - \sqrt{b}$.

**Lemma A.1.** *Let $a, b \geqslant 0$ and $D = |a - b|$. Then,*

$$\sqrt{a} \leqslant \sqrt{b} + \left( \sqrt{D} \wedge \frac{D}{2\sqrt{b}} \wedge \frac{D}{\sqrt{a}} \right). \tag{40}$$

*Proof.* Suppose $a > b$ since the bound is trivial otherwise. First, by the subadditivity of the square root, $\sqrt{a} = \sqrt{b + D} \leqslant \sqrt{b} + \sqrt{D}$. Second, $D = (\sqrt{a} - \sqrt{b})(\sqrt{a} + \sqrt{b}) \geqslant (\sqrt{a} - \sqrt{b})2\sqrt{b}$ so $\sqrt{a} \leqslant \sqrt{b} + \frac{D}{2\sqrt{b}}$. Third, $D = (\sqrt{a} - \sqrt{b})(\sqrt{a} + \sqrt{b}) \geqslant (\sqrt{a} - \sqrt{b})\sqrt{a}$ so $\sqrt{a} \leqslant \sqrt{b} + \frac{D}{\sqrt{a}}$. Taking a minimum completes the proof. $\qquad\square$

The following lemma bounds the difference of squares of two matrices.

**Lemma A.2.** *Let $\mathbf{A}, \mathbf{B}$ be two symmetric matrices. $\|\mathbf{A}^2 - \mathbf{B}^2\| \leqslant 2\|\mathbf{B}\|\|\mathbf{A} - \mathbf{B}\| + \|\mathbf{A} - \mathbf{B}\|^2$.*

*Proof.* $\|\mathbf{A}^2 - \mathbf{B}^2\| = \|(\mathbf{B} + (\mathbf{A} - \mathbf{B}))^2 - \mathbf{B}^2\| = \|\mathbf{B}(\mathbf{A} - \mathbf{B}) + (\mathbf{A} - \mathbf{B})\mathbf{B} + (\mathbf{A} - \mathbf{B})^2\| \leqslant 2\|\mathbf{B}\|\|\mathbf{A} - \mathbf{B}\| + \|\mathbf{A} - \mathbf{B}\|^2$. $\qquad\square$

The following lemma characterizes the smoothness of $\psi_{\mathrm{E}}(x) = -\log(1 - x) - x$ at 0.

**Lemma A.3.** *When $0 \leqslant x \leqslant \sqrt{2/5}$, $\psi_{\mathrm{E}}(x) \leqslant x^2$.*

*Proof.* Let $g(x) = \psi_{\mathrm{E}}(x) - x^2$. The claim follows from $g''(x) = (1 - x)^{-2} - 2 \geqslant 0$ for $x \in [0, \sqrt{2/5}]$, and $g(0) = 0$, $g(\sqrt{2/5}) < 0$. $\qquad\square$

The following *transfer rule* [Tropp, 2012, Equation 2.2] is commonly used in deriving matrix bounds.

**Lemma A.4.** *Suppose $I \subseteq \mathbb{R}$ and $f, g : I \to \mathbb{R}$ satisfies $f(x) \leqslant g(x)$, then, $f(\mathbf{X}) \preceq g(\mathbf{X})$ for any $\mathbf{X} \in \mathcal{S}_d^I$.*

It is well-known that if $X$ and $Y$ are scalar random variables such that $c \leqslant X \leqslant Y$ almost surely for some constant $c$ and that $\mathbb{E}|Y| < \infty$, it follows that $\mathbb{E}|X| < \infty$ as well, and $\mathbb{E}X \leqslant \mathbb{E}Y$. This type of "implied integrability" appears frequently in scalar concentration bounds. Let us prove its symmetric matrix extension for the sake of self-containedness.

**Lemma A.5** (Dominated integrability). *Let $\mathbf{X}$ and $\mathbf{Y}$ be $\mathcal{S}_d^{[c,\infty)}$-valued random matrices for some $c \in \mathbb{R}$ such that $\mathbf{X} \preceq \mathbf{Y}$ almost surely. Further, suppose $\mathbb{E}\mathbf{Y}$ exists. Then, so does $\mathbb{E}\mathbf{X}$ and $\mathbb{E}\mathbf{X} \preceq \mathbb{E}\mathbf{Y}$.*

*Proof.* Let us prove that each element $X_{ij}$ of the random matrix $\mathbf{X}$ is integrable. Note that for any deterministic $\mathbf{v} \in \mathbb{R}^d$, $\mathbf{v}^\mathsf{T}\mathbf{X}\mathbf{v} \leqslant \mathbf{v}^\mathsf{T}\mathbf{Y}\mathbf{v}$ almost surely. First, taking $\mathbf{v} = (0, \ldots, 0, 1, 0, \ldots 0)^\mathsf{T}$, we have

$$c \leqslant X_{jj} \leqslant Y_{jj} \quad \text{almost surely}, \tag{41}$$

concluding that the diagonal element $X_{jj}$ must be integrable (since $Y_{jj}$ is). Next, taking $\mathbf{v} = (0, \ldots, 0, 1, 0, \ldots, 0, 1, 0, \ldots, 0)^\mathsf{T}$, we have

$$2c \leqslant 2X_{ij} + X_{ii} + X_{jj} \leqslant 2Y_{ij} + Y_{ii} + Y_{jj} \quad \text{almost surely}, \tag{42}$$

concluding that $2X_{ij} + X_{ii} + X_{jj}$ must be integrable (since $2Y_{ij} + Y_{ii} + Y_{jj}$ is). Therefore, the off-diagonal element $X_{ij}$ is integrable since $X_{ii}$ and $X_{jj}$ are.

Now that we have established the existence of $\mathbb{E}\mathbf{X}$, it is clear that $\mathbb{E}\mathbf{X} \preceq \mathbb{E}\mathbf{Y}$ since for any $\mathbf{v} \in \mathbb{R}^d$, $\mathbf{v}^\mathsf{T}(\mathbb{E}\mathbf{X})\mathbf{v} = \mathbb{E}(\mathbf{v}^\mathsf{T}\mathbf{X}\mathbf{v}) \leqslant \mathbb{E}(\mathbf{v}^\mathsf{T}\mathbf{Y}\mathbf{v}) = \mathbf{v}^\mathsf{T}(\mathbb{E}\mathbf{Y})\mathbf{v}$. $\qquad\square$

## A.2 Strong Consistency of the Matrix Sample Variance

We show in this section that the matrix sample mean and variance are strongly consistent under martingale dependence, which prepares us for the upcoming proof of Corollary 4.3. First, we provide the following matrix martingale strong law of the large numbers which, we remark, holds for non-square matrices as well.

**Lemma A.6** (Matrix martingale SLLN). *Let $\mathbf{S}_n = \sum_{i=1}^n \mathbf{Z}_i$ be a matrix martingale and $\{U_n\}$ an increasing positive predictable process on $\mathcal{F}$. For any $p \in [1, 2]$, on the set*

$$\left\{ \lim_{n \to \infty} U_n = \infty, \; \sum_{n=1}^{\infty} U_n^{-p} \mathbb{E}[\mathrm{norm}^p(\mathbf{Z}_n)|\mathcal{F}_{n-1}] < \infty \right\}, \tag{43}$$

*the process $U_n^{-1}\mathbf{S}_n$ converges to 0 almost surely. Here, $\mathrm{norm}(\cdot)$ is any matrix norm.*

*Proof.* Since matrix norms are mutually equivalent, it suffices to prove the case where $\mathrm{norm}(\mathbf{A})$ is the max norm $\max_{i,j} |\mathbf{A}_{ij}|$. By the scalar martingale SLLN [Hall and Heyde, 2014, Theorem 2.18], we see that all entries of $\mathbf{S}_n$ converge to 0 almost surely. □

This immediately implies the following convergence result of the matrix sample mean and variance, where we take $\mathrm{norm}(\cdot)$ to be the usual spectral norm $\|\cdot\|$.

**Corollary A.7** (Strong consistency of the matrix sample mean). *Let $\{\mathbf{X}_n\}$ be adapted to $\mathcal{F}$ with constant conditional mean $\mathbb{E}[\mathbf{X}_n|\mathcal{F}_{n-1}] = \mathbf{M}$ and $\delta \in (0, 1]$. If $\sum_{n=1}^{\infty} n^{-1-\delta} \mathbb{E}[\|\mathbf{X}_n - \mathbf{M}\|^{1+\delta}|\mathcal{F}_{n-1}] < \infty$ almost surely, then the sample mean $\overline{\mathbf{X}}_n$ converges to $\mathbf{M}$ almost surely.*

*Proof.* It follows directly from Lemma A.6. □

The assumption $\sum_{n=1}^{\infty} n^{-1-\delta} \mathbb{E}[\|\mathbf{X}_n - \mathbf{M}\|^{1+\delta}|\mathcal{F}_{n-1}] < \infty$ is satisfied when, for example, all matrices are uniformly bounded or have a common $(1 + \delta)^{\mathrm{th}}$ conditional moment upper bound.

**Corollary A.8** (Strong consistency of the matrix sample variance). *Let $\mathbf{X}_n$ be adapted to $\mathcal{F}$ with constant conditional mean $\mathbb{E}[\mathbf{X}_n|\mathcal{F}_{n-1}] = \mathbf{M}$ and conditional variance $\mathbb{E}[(\mathbf{X}_n - \mathbf{M})^2|\mathcal{F}_{n-1}] = \mathbf{V}$. Further, assume that*

$$\sum_{n=1}^{\infty} n^{-1-\delta} \mathbb{E}[\|\mathbf{X}_n\|^{2+2\delta}|\mathcal{F}_{n-1}] < \infty \tag{44}$$

*almost surely for some $\delta \in (0, 1]$. Then, the sample variance*

$$\overline{\mathbf{V}}_n = \frac{1}{n} \sum_{i=1}^n (\mathbf{X}_i - \overline{\mathbf{X}}_n)^2 \tag{45}$$

*converges to $\mathbf{V}$ almost surely.*

*Proof.* Define $\mathbf{Q} = \mathbb{E}[\mathbf{X}_n^2|\mathcal{F}_{n-1}] = \mathbf{M}^2 + \mathbf{V}$. First, $\overline{\mathbf{X}}_n \to \mathbf{M}$ almost surely due to the constant variance with Corollary A.7. Using the inequality $\|\mathbf{A} + \mathbf{B}\|^{1+\delta} \leqslant (\|\mathbf{A}\| + \|\mathbf{B}\|)^{1+\delta} \leqslant 2^\delta(\|\mathbf{A}\|^{1+\delta} + \|\mathbf{B}\|^{1+\delta})$ (due to triangle and Jensen inequalities), we have

$$\sum_{n=1}^{\infty} n^{-1-\delta} \mathbb{E}[\|\mathbf{X}_n^2 - \mathbf{Q}\|^{1+\delta}|\mathcal{F}_{n-1}] \leqslant \sum_{n=1}^{\infty} n^{-1-\delta} 2^\delta \mathbb{E}[\|\mathbf{X}_n\|^{2+2\delta}|\mathcal{F}_{n-1}] + \sum_{n=1}^{\infty} n^{-1-\delta} 2^\delta \mathbf{Q}^{1+\delta} < \infty. \tag{46}$$

So by Corollary A.7.

$$\hat{\mathbf{Q}}_n := \frac{1}{n} \sum_{i=1}^n \mathbf{X}_i^2 \to \mathbf{Q}. \tag{47}$$

almost surely. Therefore,

$$\overline{\mathbf{V}}_n = \hat{\mathbf{Q}}_n - (\overline{\mathbf{X}}_n)^2 \to \mathbf{V} \tag{48}$$

almost surely as well, due to continuity. □

Again, the assumption $\sum_{n=1}^{\infty} n^{-1-\delta} \mathbb{E}[\|\mathbf{X}_n\|^{2+2\delta}|\mathcal{F}_{n-1}] < \infty$ is satisfied when, for example, all matrices are uniformly bounded or have a common $(2 + 2\delta)^{\mathrm{th}}$ conditional moment upper bound.

### A.3 Proof of Lemma 2.2

*Proof.* Due to the monotonicity of log, the condition (12) implies

$$\log \mathbb{E}(\exp(\mathbf{Z}_n - \mathbf{C}_n)|\mathcal{F}_{n-1}) \preceq \mathbf{C}_n'. \tag{49}$$

Now recall Lieb's concavity theorem [Lieb, 1973]: for any $\mathbf{H} \in \mathcal{S}_d$, the map $\mathbf{X} \mapsto \text{tr} \exp(\mathbf{H} + \log \mathbf{X})$ ($\mathcal{S}_d^{++} \to (0, \infty)$) is concave. Therefore,

$$\mathbb{E}(L_n|\mathcal{F}_{n-1}) = \mathbb{E}\left(\text{tr} \exp\left(\sum_{i=1}^{n-1} \mathbf{Z}_i - \sum_{i=1}^{n-1}(\mathbf{C}_i + \mathbf{C}_i') - \mathbf{C}_n' + \log e^{\mathbf{Z}_n - \mathbf{C}_n}\right)\middle|\mathcal{F}_{n-1}\right) \tag{50}$$

(Jensen's inequality)

$$\leqslant \text{tr} \exp\left(\sum_{i=1}^{n-1} \mathbf{Z}_i - \sum_{i=1}^{n-1}(\mathbf{C}_i + \mathbf{C}_i') - \mathbf{C}_n' + \log \mathbb{E}(e^{\mathbf{Z}_n - \mathbf{C}_n}|\mathcal{F}_{n-1})\right) \tag{51}$$

(by (49) and monotonicity of trace)

$$\leqslant \text{tr} \exp\left(\sum_{i=1}^{n-1} \mathbf{Z}_i - \sum_{i=1}^{n-1}(\mathbf{C}_i + \mathbf{C}_i') - \mathbf{C}_n' + \mathbf{C}_n'\right) = L_{n-1}, \tag{52}$$

concluding the proof that $\{L_n\}$ is a supermartingale. Finally, observe that

$$L_n = \text{tr} \exp\left(\sum_{i=1}^{n} \mathbf{Z}_i - \sum_{i=1}^{n}(\mathbf{C}_i + \mathbf{C}_i')\right) \tag{53}$$

$$\geqslant \text{tr} \exp\left(\sum_{i=1}^{n} \mathbf{Z}_i - \lambda_{\max}\left(\sum_{i=1}^{n}(\mathbf{C}_i + \mathbf{C}_i')\right)\mathbf{I}\right) \tag{54}$$

$$\geqslant \lambda_{\max} \exp\left(\sum_{i=1}^{n} \mathbf{Z}_i - \lambda_{\max}\left(\sum_{i=1}^{n}(\mathbf{C}_i + \mathbf{C}_i')\right)\mathbf{I}\right) \tag{55}$$

$$= \exp \lambda_{\max}\left(\sum_{i=1}^{n} \mathbf{Z}_i - \lambda_{\max}\left(\sum_{i=1}^{n}(\mathbf{C}_i + \mathbf{C}_i')\right)\mathbf{I}\right) \tag{56}$$

$$= \exp\left(\lambda_{\max}\left(\sum_{i=1}^{n} \mathbf{Z}_i\right) - \lambda_{\max}\left(\sum_{i=1}^{n}(\mathbf{C}_i + \mathbf{C}_i')\right)\right), \tag{57}$$

concluding the proof. $\qquad\square$

### A.4 Proof of Theorem 3.1

*Proof.* Note that $2\mathbf{V}_n^*$ is the sample average of $n/2$ independent random matrices, each with eigenvalues in $[0, 1]$, of common mean $2\mathbf{V}$ and second moment upper bound

$$\|\mathbb{E}(\mathbf{X}_1 - \mathbf{X}_2)^4\| \leqslant \|\mathbb{E}(\mathbf{X}_1 - \mathbf{X}_2)^2\| = 2\|\mathbf{V}\|. \tag{58}$$

Thus applying the Matrix Bennett-Bernstein inequality (5) on $2\mathbf{V}_n^*$, we see that

$$\mathbb{P}\left(\|2\mathbf{V}_n^* - 2\mathbf{V}\| \geqslant \frac{2\log(2d/\alpha)}{3n} + \sqrt{\frac{8\log(2d/\alpha)\|\mathbf{V}\|}{n}}\right) \leqslant \alpha. \tag{59}$$

Therefore, with probability at least $1 - \alpha$,

$$\big|\, \|\mathbf{V}_n^*\| - \|\mathbf{V}\|\, \big| \leqslant \|\mathbf{V}_n^* - \mathbf{V}\| \leqslant \frac{\log(2d/\alpha)}{3n} + \sqrt{\frac{2\log(2d/\alpha)\|\mathbf{V}\|}{n}}. \tag{60}$$

Denote by $g$ the constant $\sqrt{\frac{5}{6}} + \frac{1}{\sqrt{2}}$, which satisfies $\frac{1}{3g} + \sqrt{2} = g$. The above implies that if $\|\mathbf{V}\| \geqslant \frac{g^2 \log(2d/\alpha)}{n}$,

$$\big|\, \|\mathbf{V}_n^*\| - \|\mathbf{V}\|\, \big| \leqslant \sqrt{\frac{\log(2d/\alpha)\|\mathbf{V}\|}{9g^2 n}} + \sqrt{\frac{2\log(2d/\alpha)\|\mathbf{V}\|}{n}} = \sqrt{\frac{g^2 \log(2d/\alpha)\|\mathbf{V}\|}{n}}, \tag{61}$$

which in turn implies that, via Lemma A.1,

$$\|\mathbf{V}\|^{1/2} \leqslant \|\mathbf{V}_n^*\|^{1/2} + \sqrt{\frac{g^2 \log(2d/\alpha)}{n}}. \tag{62}$$

Since the above also holds if $\|\mathbf{V}\| < \frac{g^2 \log(2d/\alpha)}{n}$, we see that the inequality above holds with probability at least $1 - \alpha$ regardless of the true value of $\|\mathbf{V}\|$. Integrating the inequality above into the matrix Bennett-Bernstein inequality (5) via an $\alpha = \alpha(n-1)/n + \alpha/n$ union bound, we arrive at

$$\lambda_{\max}\left(\overline{\mathbf{X}}_n - \mathbf{M}\right) \leqslant \frac{\log \frac{nd}{(n-1)\alpha}}{3n} + \sqrt{\frac{2 \log \frac{nd}{(n-1)\alpha}}{n}} \left(\|\mathbf{V}_n^*\|^{1/2} + \left(\sqrt{\frac{5}{6}} + \frac{1}{\sqrt{2}}\right)\sqrt{\frac{\log(2nd/\alpha)}{n}}\right) \tag{63}$$

with probability at least $1 - \alpha$. This concludes the proof. The asymptotics (18) follows simply from the strong law of large numbers and the continuity of the spectral norm. $\qquad\square$

### A.5 Proof of Lemma 4.1

*Proof.* Recall that $\psi_{\mathrm{E}}(\gamma) = -\log(1 - \gamma) - \gamma$. An inequality by Fan et al. [2015] quoted by Howard et al. [2021, Appendix A.8] states that, for all $0 \leqslant \gamma < 1$ and $\xi \geqslant -1$,

$$\exp(\gamma\xi - \psi_{\mathrm{E}}(\gamma)\xi^2) \leqslant 1 + \gamma\xi. \tag{64}$$

Since $\mathbf{X}_n - \widehat{\mathbf{X}}_n \in \mathcal{S}_d^{[-1,\infty)}$, we can apply the transfer rule (Lemma A.4), replacing the scalar $\xi$ above by the matrix $\mathbf{X}_n - \widehat{\mathbf{X}}_n$, and plugging in $\gamma = \gamma_n \in (0, 1)$,

$$\exp(\gamma_n(\mathbf{X}_n - \widehat{\mathbf{X}}_n) - \psi_{\mathrm{E}}(\gamma_n)(\mathbf{X}_n - \widehat{\mathbf{X}}_n)^2) \preceq 1 + \gamma_n(\mathbf{X}_n - \widehat{\mathbf{X}}_n). \tag{65}$$

Lemma A.5 then guarantees the integrability of the left hand side, and that

$$\mathbb{E}\left(\exp(\gamma_n(\mathbf{X}_n - \widehat{\mathbf{X}}_n) - \psi_{\mathrm{E}}(\gamma_n)(\mathbf{X}_n - \widehat{\mathbf{X}}_n)^2)\Big|\mathcal{F}_{n-1}\right) \preceq \mathbb{E}\left(1 + \gamma_n(\mathbf{X}_n - \widehat{\mathbf{X}}_n)\Big|\mathcal{F}_{n-1}\right) \tag{66}$$

$$= 1 + \gamma_n(\mathbf{M}_n - \widehat{\mathbf{X}}_n) \preceq \exp(\gamma_n(\mathbf{M}_n - \widehat{\mathbf{X}}_n)), \tag{67}$$

where in the final step we use the transfer rule again with $1 + x \leqslant \exp(x)$ for all $x \in \mathbb{R}$. This concludes the proof. $\qquad\square$

### A.6 Proof of Corollary 4.3

*Proof.* First, it is straightforward that $\lambda_{\max}(\mathbf{X}_i - \overline{\mathbf{X}}_{i-1}) \geqslant -1$ for every $i = 1, \ldots, n$ since both $\mathbf{X}_i$ and $\overline{\mathbf{X}}_{i-1}$ take values in $\mathcal{S}_d^{[0,1]}$, so Theorem 4.2 is applicable. Let us prove the two claims about the deviation bound $D_n^{\mathsf{meb2}}$ under $\gamma_i = \sqrt{\frac{2 \log(d/\alpha)}{n \overline{v}_{i-1}}}$. Recall that

$$\overline{\mathbf{V}}_0 = \frac{1}{4}\mathbf{I}, \quad \overline{\mathbf{V}}_k = \frac{1}{k}\sum_{i=1}^k (\mathbf{X}_i - \overline{\mathbf{X}}_k)^2, \quad \overline{v}_k = \|\overline{\mathbf{V}}_k\| \vee \frac{5 \log(d/\alpha)}{n}, \tag{68}$$

$$\widetilde{s}_n = \lambda_{\max}\left(\frac{1}{n}\sum_{i=1}^n \frac{(\mathbf{X}_i - \overline{\mathbf{X}}_{i-1})^2}{\overline{v}_{i-1}}\right). \tag{69}$$

Let us compute the almost sure limit $\lim_{n \to \infty} \sqrt{n} D_n^{\mathsf{meb2}}$. First, the following limits hold almost surely due to Corollaries A.7 and A.8:

$$\lim_{k \to \infty} \overline{\mathbf{X}}_k = \mathbf{M}, \quad \lim_{k \to \infty} \overline{\mathbf{V}}_k = \mathbf{V}, \quad \lim_{k \to \infty} \overline{v}_k = \|\mathbf{V}\|. \tag{70}$$

Let us compute the limit

$$\lim_{n \to \infty} \frac{1}{n}\sum_{i=1}^n \frac{(\mathbf{X}_i - \overline{\mathbf{X}}_{i-1})^2}{\overline{v}_{i-1}}. \tag{71}$$

To do this, we observe that

$$\lim_{n \to \infty} \frac{1}{n}\sum_{i=1}^n \frac{(\mathbf{X}_i - \mathbf{M})^2}{\|\mathbf{V}\|} = \frac{\mathbf{V}}{\|\mathbf{V}\|} \tag{72}$$

almost surely due to Corollary A.7. Bounding the difference

$$\left\| \frac{1}{n} \sum_{i=1}^{n} \frac{(\mathbf{X}_i - \overline{\mathbf{X}}_{i-1})^2}{\overline{v}_{i-1}} - \frac{1}{n} \sum_{i=1}^{n} \frac{(\mathbf{X}_i - \mathbf{M})^2}{\|\mathbf{V}\|} \right\| \tag{73}$$

$$\leqslant \frac{1}{n} \sum_{i=1}^{n} \left( \frac{\|(\mathbf{X}_i - \overline{\mathbf{X}}_{i-1})^2 - (\mathbf{X}_i - \mathbf{M})^2\|}{\overline{v}_{i-1}} + \|\mathbf{X}_i - \mathbf{M}\|^2 |\overline{v}_{i-1}^{-1} - \|\mathbf{V}\|^{-1}| \right) \tag{74}$$

(Lemma A.2) $\tag{75}$

$$\leqslant \frac{1}{n} \sum_{i=1}^{n} \left( \frac{2\|\mathbf{X}_i - \mathbf{M}\|\|\overline{\mathbf{X}}_{i-1} - \mathbf{M}\| + \|\overline{\mathbf{X}}_{i-1} - \mathbf{M}\|^2}{\overline{v}_{i-1}} + \|\mathbf{X}_i - \mathbf{M}\|^2 |\overline{v}_{i-1}^{-1} - \|\mathbf{V}\|^{-1}| \right) \tag{76}$$

$$(\|\overline{\mathbf{X}}_{i-1} - \mathbf{M}\| \leqslant 2, \|\mathbf{X}_i - \mathbf{M}\| \leqslant 2) \tag{77}$$

$$\leqslant \frac{1}{n} \sum_{i=1}^{n} \frac{6\|\overline{\mathbf{X}}_{i-1} - \mathbf{M}\|}{\overline{v}_{i-1}} + \frac{1}{n} \sum_{i=1}^{n} 4|\overline{v}_{i-1}^{-1} - \|\mathbf{V}\|^{-1}| \to 0. \tag{78}$$

Therefore, we see that

$$\frac{1}{n} \sum_{i=1}^{n} \frac{(\mathbf{X}_i - \overline{\mathbf{X}}_{i-1})^2}{\overline{v}_{i-1}} \to \frac{\mathbf{V}}{\|\mathbf{V}\|}, \quad \widetilde{s}_n = \lambda_{\max} \left( \frac{1}{n} \sum_{i=1}^{n} \frac{(\mathbf{X}_i - \overline{\mathbf{X}}_{i-1})^2}{\overline{v}_{i-1}} \right) \to 1 \tag{79}$$

almost surely.

We can now use the expansion $\psi_{\mathrm{E}}(x) = \sum_{k=2}^{\infty} \frac{x^k}{k}$ to obtain,

$$\limsup_{n \to \infty} \sqrt{n} D_n^{\mathsf{meb2}} \tag{80}$$

$$= \limsup_{n \to \infty} \frac{\log(d/\alpha) + \lambda_{\max} \left( \sum_{i=1}^{n} \psi_{\mathrm{E}} \left( \sqrt{\frac{2\log(d/\alpha)}{n\overline{v}_{i-1}}} \right) (\mathbf{X}_i - \overline{\mathbf{X}}_{i-1})^2 \right)}{\frac{1}{n} \sum_{i=1}^{n} \sqrt{\frac{2\log(d/\alpha)}{\overline{v}_{i-1}}}} \tag{81}$$

$$\leqslant \limsup_{n \to \infty} \frac{\log(d/\alpha) + \lambda_{\max} \left( \sum_{i=1}^{n} \frac{1}{2} \left( \sqrt{\frac{2\log(d/\alpha)}{n\overline{v}_{i-1}}} \right)^2 (\mathbf{X}_i - \overline{\mathbf{X}}_{i-1})^2 \right)}{\frac{1}{n} \sum_{i=1}^{n} \sqrt{\frac{2\log(d/\alpha)}{\overline{v}_{i-1}}}} \tag{82}$$

$$+ \sum_{k=3}^{\infty} \limsup_{n \to \infty} \underbrace{\frac{\lambda_{\max} \left( \sum_{i=1}^{n} \frac{1}{k} \left( \sqrt{\frac{2\log(d/\alpha)}{n\overline{v}_{i-1}}} \right)^k (\mathbf{X}_i - \overline{\mathbf{X}}_{i-1})^2 \right)}{\frac{1}{n} \sum_{i=1}^{n} \sqrt{\frac{2\log(d/\alpha)}{\overline{v}_{i-1}}}}}_{T_{k,n}}. \tag{83}$$

Let us prove that $\lim_{n \to \infty} T_{k,n} = 0$ for all $k \geqslant 3$. To see that,

$$T_{k,n} \leqslant \frac{\sum_{i=1}^{n} \frac{1}{k} \left( \sqrt{\frac{2\log(d/\alpha)}{n\overline{v}_{i-1}}} \right)^k \|\mathbf{X}_i - \overline{\mathbf{X}}_{i-1}\|^2}{\frac{1}{n} \sum_{i=1}^{n} \sqrt{\frac{2\log(d/\alpha)}{\overline{v}_{i-1}}}} \leqslant \frac{\sum_{i=1}^{n} \frac{1}{k} \left( \sqrt{\frac{2\log(d/\alpha)}{n\overline{v}_{i-1}}} \right)^k}{\frac{1}{n} \sum_{i=1}^{n} \sqrt{\frac{2\log(d/\alpha)}{\overline{v}_{i-1}}}} \tag{84}$$

$$= \frac{k^{-1} \left( 2\log(d/\alpha) \right)^{\frac{k-1}{2}} n^{-\frac{k-2}{2}} \boxed{n^{-1} \sum_{i=1}^{n} \overline{v}_{i-1}^{-k/2}}}{\boxed{n^{-1} \sum_{i=1}^{n} \overline{v}_{i-1}^{-1/2}}}. \tag{85}$$

Since the boxed terms converge to non-zero quantities, we see that $T_{k,n}$ converges to 0 due to the $n^{-\frac{k-2}{2}}$ term. Therefore,

$$\limsup_{n \to \infty} \sqrt{n} D_n^{\mathsf{meb2}} \tag{86}$$

$$\leqslant \limsup_{n \to \infty} \frac{\log(d/\alpha) + \lambda_{\max}\left(\sum_{i=1}^{n} \frac{1}{2} \left(\sqrt{\frac{2\log(d/\alpha)}{n \overline{v}_{i-1}}}\right)^2 (\mathbf{X}_i - \overline{\mathbf{X}}_{i-1})^2\right)}{\frac{1}{n} \sum_{i=1}^{n} \sqrt{\frac{2\log(d/\alpha)}{\overline{v}_{i-1}}}} \tag{87}$$

$$= \limsup_{n \to \infty} \sqrt{\frac{\log(d/\alpha)}{2}} \frac{\left(1 + \lambda_{\max}\left(\frac{1}{n} \sum_{i=1}^{n} \frac{(\mathbf{X}_i - \overline{\mathbf{X}}_{i-1})^2}{\overline{v}_{i-1}}\right)\right)}{\frac{1}{n} \sum_{i=1}^{n} \overline{v}_{i-1}^{-1/2}} \tag{88}$$

$$= \sqrt{2\log(d/\alpha)\|\mathbf{V}\|}. \tag{89}$$

Similarly, one can show that $\liminf_{n \to \infty} \sqrt{n} D_n^{\mathsf{meb2}} \geqslant \sqrt{2\log(d/\alpha)\|\mathbf{V}\|}$, concluding the proof. We remark that the proof above strengthens that of Waudby-Smith and Ramdas [2023, Lemmas 4-8]. $\square$

# B  Additional Concentration Inequalities

## B.1  Remarks on the Scalar (1) and Matrix (5) Bennett-Bernstein Inequalities

Non-empirical Bernstein inequalities are typically stated in terms of the upper bound of the tail probability $\mathbb{P}(S_n - \mathbb{E}S_n \geqslant t)$. These are derived via Bennett-type inequalities via controlling the function

$$h(u) = (1 + u)\log(1 + u) - u \overset{(*)}{\geqslant} \frac{u^2}{2(1 + u/3)}. \tag{90}$$

We, for statistical purposes however, are interested in deviation bounds under a fixed error probability $\alpha$. The Bennett-to-Bernstein conversion (*) is looser than the following inequality.

**Lemma B.1.** *For all $x \geqslant 0$, $h^{-1}(x) \leqslant \sqrt{2x} + x/3$.*

A proof of this polynomial upper bound on $h^{-1}$ can be found from Equation (45) onwards in Audibert et al. [2009]. Tropp [2012, Theorem 6.1] first states a matrix Bennett bound in terms of the $h$ function, then uses (*) to obtain a closed-formed matrix Bernstein bound, both controlling the tail probability $\mathbb{P}(\lambda_{\max}(\mathbf{S}_n - \mathbb{E}\mathbf{S}_n) \geqslant t)$. Let us use Lemma B.1 to recover a fixed-error $\alpha$ bound whose tightness is between the matrix Bennett and the matrix Bernstein, which we already recorded in the paper as (5).

**Proposition B.2** (Matrix Bennett-Bernstein inequality (5))**.** *Let $\overline{\mathbf{X}}_n$ be the sample average of independent $d \times d$ symmetric matrices $\mathbf{X}_1, \ldots, \mathbf{X}_n$ with common mean $\mathbb{E}\mathbf{X}_i = \mathbf{M}$, common eigenvalue upper bound $\lambda_{\max}(\mathbf{X}_i) \leqslant B$, and $\sum_{i=1}^{n} \mathbb{E}\mathbf{X}_i^2 \preceq n\mathbf{V}$. For any $\alpha > 0$*

$$\mathbb{P}\left(\lambda_{\max}\left(\overline{\mathbf{X}}_n - \mathbf{M}\right) \geqslant \frac{B\log(d/\alpha)}{3n} + \sqrt{\frac{2\log(d/\alpha)\|\mathbf{V}\|}{n}}\right) \leqslant \alpha. \tag{91}$$

*Proof.* Due to Tropp [2012, Equation (i) in Proof of Theorem 6.1],

$$\mathbb{P}\left[\lambda_{\max}\left(\overline{\mathbf{X}}_n - \mathbf{M}\right) \geqslant t\right] \leqslant d \cdot \exp\left(-\frac{n\lambda_{\max}(\mathbf{V})}{B^2} \cdot h\left(\frac{Bt}{\lambda_{\max}(\mathbf{V})}\right)\right). \tag{92}$$

Setting the right hand side as $\alpha$, we obtain via Lemma B.1

$$t = \frac{\lambda_{\max}(\mathbf{V})}{B} h^{-1}\left(\frac{\log(d/\alpha)B^2}{n\lambda_{\max}(\mathbf{V})}\right) \leqslant \frac{\lambda_{\max}(\mathbf{V})}{B}\left(\sqrt{\frac{2\log(d/\alpha)B^2}{n\lambda_{\max}(\mathbf{V})}} + \frac{\log(d/\alpha)B^2}{3n\lambda_{\max}(\mathbf{V})}\right), \tag{93}$$

which readily leads to the bound (5)

$$\mathbb{P}\left(\lambda_{\max}\left(\overline{\mathbf{X}}_n - \mathbf{M}\right) \geqslant \frac{B\log(d/\alpha)}{3n} + \sqrt{\frac{2\log(d/\alpha)\lambda_{\max}(\mathbf{V})}{n}}\right) \leqslant \alpha. \tag{94}$$

We also remark that the scalar case (1) is when $d = 1$. $\square$

## B.2 Sharp Maurer-Pontil Inequality

Maurer and Pontil [2009, Theorem 4] derived a scalar empirical Bernstein inequality which we quote as (2), by a union bound between a scalar Bennett-Bernstein inequality and a tail bound on the sample variance. However, their balanced union bound split $\alpha = \alpha/2 + \alpha/2$ leads to the looser $\log(2/\alpha)$ term. This causes the confidence interval to be 10.9675% longer when $\alpha = 0.05$ in the large sample limit. We slightly modify their proof below to obtain a sharp EB inequality for scalars.

**Proposition B.3.** *Let $X_1, \ldots, X_n$ be $[0, 1]$-bounded independent random scalars with common mean $\mu$ and variance $\sigma^2$. We denote by $\overline{X}_n$ their sample average and $\hat{\sigma}_n^2$ the Bessel-corrected sample variance. Then, for any $\alpha \in (0, 1)$, $\mathbb{P}\left(\overline{X}_n - \mu \geqslant \rho_n\right) \leqslant \alpha$, where*

$$\rho_n = \frac{\log \frac{n}{(n-1)\alpha}}{3n} + \sqrt{\frac{2\hat{\sigma}_n^2 \log \frac{n}{(n-1)\alpha}}{n}} + 2\sqrt{\frac{\left(\log \frac{n}{(n-1)\alpha}\right)\left(\log \frac{n}{\alpha}\right)}{n(n-1)}}. \tag{95}$$

*Further, with i.i.d. $X_1, \ldots, X_n$,*

$$\lim_{n \to \infty} \sqrt{n}\rho_n = \sqrt{2\sigma^2 \log(1/\alpha)}, \quad \text{almost surely.} \tag{96}$$

*Proof.* By Bennett-Bernstein inequality (1),

$$\mathbb{P}\left(\overline{X}_n - \mu \geqslant \frac{\log(1/\alpha)}{3n} + \sqrt{\frac{2\sigma^2 \log(1/\alpha)}{n}}\right) \leqslant \alpha. \tag{97}$$

The deviation of $\hat{\sigma}_n^2$ from $\sigma^2$ is controlled by a self-bounding concentration inequality [Maurer and Pontil, 2009, Theorem 7],

$$\mathbb{P}\left(\sigma - \hat{\sigma}_n \geqslant \sqrt{\frac{2\log(1/\alpha)}{n-1}}\right) \leqslant \alpha. \tag{98}$$

The desired bound thus follows from an $\alpha = \alpha(n-1)/n + \alpha/n$ union bound. $\square$

| Sample Size ($n$) | Original MP | Sharp MP | Self-normalized |
|:---:|:---:|:---:|:---:|
| 100 | 2.120 | 2.256 | 1.527 |
| 1,000 | 1.441 | 1.476 | 1.081 |
| 10,000 | 1.214 | 1.169 | 1.017 |
| 100,000 | 1.144 | 1.059 | 1.005 |
| 1,000,000 | 1.121 | 1.021 | 1.002 |

Table 2: Lengths of 95% one-sided confidence intervals of three empirical Bernstein inequalities divided by that of the (oracle) Bennett-Bernstein inequality (1). "Original MP" stands for the result of Maurer and Pontil [2009, Theorem 4]; "Sharp MP" our sharpened result Proposition B.3; and "Self-normalized" the result by Waudby-Smith and Ramdas [2023, Theorem 2, Remark 1].

As can be seen from the simulation with $\mathrm{Unif}_{[0,1]}$-distributed random variables displayed in Table 2, our sharpened EB inequality leads to significant improvement for large samples. Still, the EB inequality based on the self-normalized technique by Howard et al. [2021], Waudby-Smith and Ramdas [2023] has a much smaller gap compared to the oracle Bernstein inequality.

## B.3 First Matrix EB Inequality with the Classical Sample Variance

Recall in Section 3 we proved the first matrix EB inequality by applying the matrix Bennett-Bernstein inequality (5) twice, once on the sample average and once on the paired sample variance $\mathbf{V}_n^*$. The latter step differs from the scalar result by Maurer and Pontil [2009] which uses a self-bounding technique on the classical Bessel-corrected sample variance. Intuitively however, the classical matrix Bessel-corrected sample variance for $\mathbf{X}_1, \ldots, \mathbf{X}_n$

$$\widehat{\mathbf{V}}_n = \frac{1}{n(n-1)} \sum_{1 \leqslant i < j \leqslant n} (\mathbf{X}_i - \mathbf{X}_j)^2, \tag{99}$$

should have a better convergence rate to the population variance $\mathbf{V}$ than $\mathbf{V}_n^*$ as a full U-statistic. This fact is captured by the self-bounding concentration inequality Maurer and Pontil [2009] employ in the scalar case. For matrices, however, the best analysis of $\widehat{\mathbf{V}}_n$ we are aware of comes from the Efron-Stein technique due to Paulin et al. [2016], which leads to the following variant of Theorem 3.1.

**Theorem B.4** (First matrix empirical Bernstein inequality, classical sample variance)**.** *Let* $\mathbf{X}_1, \ldots, \mathbf{X}_n$ *be* $\mathcal{S}_d^{[0,1]}$*-valued independent random matrices with common mean* $\mathbf{M}$ *and variance* $\mathbf{V}$*. We denote by* $\overline{\mathbf{X}}_n$ *their sample average and* $\widehat{\mathbf{V}}_n$ *the Bessel-corrected sample variance. Then, for any* $\alpha \in (0, 1)$,

$$\mathbb{P}\left( \lambda_{\max}\left( \overline{\mathbf{X}}_n - \mathbf{M} \right) \geqslant D_n^{\mathsf{meb1c}} \right) \leqslant \alpha, \tag{100}$$

*where*

$$D_n^{\mathsf{meb1c}} = \sqrt{\frac{2 \log \frac{nd}{(n-1)\alpha}}{n}} \left( \|\widehat{\mathbf{V}}_n\|^{1/2} + \sqrt{\frac{\log(2nd/\alpha)}{2n\|\widehat{\mathbf{V}}_n\|}} \wedge \left( \frac{2 \log(2nd/\alpha)}{n} \right)^{1/4} \right) + \frac{\log \frac{nd}{(n-1)\alpha}}{3n}. \tag{101}$$

*Further, if* $\mathbf{X}_1, \ldots, \mathbf{X}_n$ *are i.i.d.,*

$$\lim_{n \to \infty} \sqrt{n} D_n^{\mathsf{meb1c}} = \sqrt{2 \log(d/\alpha)\|\mathbf{V}\|}, \quad \text{almost surely.} \tag{102}$$

*Proof.* We view the classical sample variance $\widehat{\mathbf{V}}_n \in \mathcal{S}_d^{[0,1]}$ as a bounded matrix function of independent variables $\mathbf{X}_1, \ldots, \mathbf{X}_n$. Let $\widehat{\mathbf{V}}_n^j$ be the sample variance by replacing $\mathbf{X}_j$ with an i.i.d. copy $\mathbf{X}_j'$. The Efron-Stein variance proxy of $\widehat{\mathbf{V}}_n$ satisfies

$$\frac{1}{2} \sum_{j=1}^n \mathbb{E}[(\widehat{\mathbf{V}}_n - \widehat{\mathbf{V}}_n^j)^2 | \mathbf{X}_1, \ldots, \mathbf{X}_n] \preceq \frac{1}{2n} \mathbf{I}, \tag{103}$$

which can be noted from the fact that each $\widehat{\mathbf{V}}_n - \widehat{\mathbf{V}}_n^j \in \mathcal{S}_d^{[-1/n, 1/n]}$. We now invoke the Efron-Stein tail bound, Corollary 5.1 from Paulin et al. [2016] to see that for any $t > 0$,

$$\mathbb{P}(| \|\mathbf{V}\| - \|\widehat{\mathbf{V}}_n\| | \geqslant t) \leqslant \mathbb{P}(\|\mathbf{V} - \widehat{\mathbf{V}}_n\| \geqslant t) \leqslant 2d \exp\left( \frac{-nt^2}{2} \right). \tag{104}$$

Setting the right hand side to $\alpha$, we obtain, with probability at least $1 - \alpha$,

$$| \|\mathbf{V}\| - \|\widehat{\mathbf{V}}_n\| | < \sqrt{\frac{2 \log(2d/\alpha)}{n}}, \tag{105}$$

which, due to Lemma A.1, implies that

$$\|\mathbf{V}\|^{1/2} < \|\widehat{\mathbf{V}}_n\|^{1/2} + \sqrt{\frac{\log(2d/\alpha)}{2n\|\widehat{\mathbf{V}}_n\|}} \wedge \left( \frac{2 \log(2d/\alpha)}{n} \right)^{1/4}. \tag{106}$$

Integrating the inequality above into the matrix Bennett-Bernstein inequality (5) via an $\alpha = \alpha(n - 1)/n + \alpha/n$ union bound concludes the proof. The asymptotics (102) follows simply from the strong consistency of the sample variance and the continuity of the spectral norm. $\square$

We extend the previous simulation shown in Table 1 with this additional matrix EB inequality. As can be seen from Table 3, using the sample variance leads to a slightly larger confidence set despite the asymptotics (102) still being sharp. We leave it to future work whether an improved analysis of the classical matrix sample variance better than (105) can lead to a Maurer-Pontil-style matrix EB inequality that beats Theorem 3.1.

## C  Other Scalar EB Inequalities

A wealth of scalar empirical Bernstein inequalities exists in the learning theory literature, especially under the PAC-Bayes [Alquier, 2024] setup. These are concentration inequalities on the posterior expected deviation, valid simultaneously over all posterior distributions. We provide a brief bibliographical remark in this section.

| Sample Size ($n$) | $D_n^{\text{meb1}}/D_n^{\text{tb}}$ | $D_n^{\text{meb1c}}/D_n^{\text{tb}}$ | $D_n^{\text{meb2}}/D_n^{\text{tb}}$ |
|---|---|---|---|
| 100 | 2.612 | 3.057 | 1.397 |
| 1,000 | 1.589 | 1.874 | 1.105 |
| 10,000 | 1.214 | 1.313 | 1.022 |
| 100,000 | 1.072 | 1.109 | 1.007 |
| 1,000,000 | 1.024 | 1.037 | 1.002 |

Table 3: Relative lengths of 95% one-sided confidence sets by three sharp matrix empirical Bernstein inequalities compared to the (oracle) matrix Bennett-Bernstein inequality (5). This table includes an additional column for $D_n^{\text{meb1c}}$ on top of Table 1.

A PAC-Bayes EB inequality was proved by Tolstikhin and Seldin [2013] using a technique comparable to Audibert et al. [2009], Maurer and Pontil [2009]: they used a union bound to combine an oracle Bernstein-style PAC-Bayes bound and a sample variance PAC-Bayes bound. When taking a degenerate parameter space (i.e. singleton prior and posterior distributions), one recovers an EB inequality for $[0, 1]$-bounded i.i.d. random variables, with a larger constant on the $n^{-1/2}$ term compared to (2), therefore is not sharp by our standard. An alternative to EB inequalities called the "un-expected Bernstein inequality" was introduced by Mhammedi et al. [2019], which "together with the standard Bernstein inequality imply [*sic*] a version of the empirical Bernstein inequality with slightly worse factors" [Mhammedi et al., 2019, Appendix G]. A much tighter, time-uniform PAC-Bayes EB inequality was later proved by Jang et al. [2023, Corollary 4], much similar to the "betting" technique of Waudby-Smith and Ramdas [2023]. However, no sharp fixed-time EB inequality was proved by Jang et al. [2023]. A similar PAC-Bayes time-uniform result was obtained by Mhammedi [2021, Theorem 6], also suffering from non-sharpness when instantiated fixed-time.

We finally quote a recent study by Shekhar and Ramdas [2023] that compares various scalar EB inequalities via their first- and second-order expansions. They show that a class of betting-driven confidence intervals, including those proposed by Waudby-Smith and Ramdas [2023] (other than (3)) and the recent universal portfolio-based bound by Orabona and Jun [2024], have a minimax optimal rate that outperforms even the sharpness criterion (4) for scalar EB inequalities. However, these confidence sets are not in closed form, and it remains an open problem (even in the scalar case) if there are closed-form empirical inequalities that achieve so.

