# OpenReview forum: "Sharp Matrix Empirical Bernstein Inequalities"
_NeurIPS.cc/2025/Conference — NeurIPS 2025 poster_

### Official Review · Reviewer_ZodG · 2025-06-18

**Clarity:** 4
**Significance:** 3
**Originality:** 3
**Rating:** 5
**Confidence:** 3

**Summary:**

The paper proves two empirical Bernstein inequalities for sums of symmetric random matrices, extending the empirical Bernstein bounds from the scalar to the matrix setting. Suppose $T_\alpha$ satisfies $\Pr(\lambda_{\max}(\sum_i X_i) \geq T_\alpha) \leq \alpha$, where $X_i$'s are mean-zero symmetric matrices. The standard (non-empirical) matrix Bernstein inequality gives
$$
T_\alpha = \sqrt{\frac{2\log(d/\alpha) \|V\|}{n}} + O\left(\frac{\log(1/\alpha)}{n}\right),
$$
where $V = \frac{1}{n} \sum_i \mathbb{E} X_i^2$, and each $-I \preceq X_i \preceq I$. In the empirical versions, the unknown variance $V$ is replaced with a data-dependent estimator.

Specifically, the paper shows show that
$$
T_\alpha = \sqrt{\frac{2 \log(nd/((n-1)\alpha)) \Vert V_n^\ast \Vert }{n}} + O\left( \frac{\log(nd/\alpha)}{n} \right),
$$
where $V_n^* = \frac{1}{n} \sum_{i\ \text{odd}} (X_i - X_{i+1})^2$ is the paired sample variance. This result is shown to be sharp asymptotically. The sharpness is obtained through an unbalanced union bound, improving on the scalar case of Maurer and Pontil.

The paper also proves a second empirical Bernstein inequality under martingale dependence, generalizing the classical matrix Freedman inequality. The technique is based upon the self-normalized supermartingales due to Waudby-Smith and Ramdas (2023) but generalized to the matrix case.

**Questions:**

Can you comment more on Kroshnin and Suvorikova’s result? Their result contains an \hat{z} term but that is just 1 in many cases. If you only want a bound that is tight in order (without the \hat{z} term in Kroshnin and Suvorikova’s result) instead of having sharp constants, is there a simpler proof?

**Ethical Concerns:**

["NO or VERY MINOR ethics concerns only"]

**Limitations:**

Yes

**Paper Formatting Concerns:**

NIL

**Quality:**

4

**Strengths And Weaknesses:**

Strengths: The paper generalizes the classical matrix Bernstein/Freedman inequalities to the empirical case and obtains sharp bounds.

Weaknesses: Previous work of Kroshnin and Suvorikova seems to have a good bound with much simpler proofs. I do not find this paper's improvement very important in applications (unless one cares a lot about the constants).

---

> ### Author Rebuttal · Authors · 2025-07-30
>
> ## More comment on Kroshnin and Suvorikova
>
> Their EB inequality is not "sharp" by our definition in Eq.(8). Just as the scalar Maurer-Pontil bound can be sharped by an imbalance $\alpha$ split (Appendix B.2), the KS inequality can be similarly sharpened.
>
> However, even sharpened, the bound (as the reviewer aptly pointed out) suffers from a large constant of 15 in its lower-order term.
>
> To us, the main insight from KS is its analysis (Lemma 4.9) on the vanilla matrix sample variance (unlike our paired one). It exhibits another facet of the trade-off together with our appendicized analysis of the vanilla sample variance (Eqn.(105)): they avoid the $n^{-3/4}$ term at the price of a large constant in the $n^{-1}$ term.
>
> Still, as we mentioned, we still hope future work to address "whether an improved analysis of the classical matrix sample variance better than (104) can lead to a Maurer-Pontil-style matrix EB inequality that beats Theorem 3.1."
>
> We finally note that the KS paper was concurrent work to ours, using complementary techniques (and with different goals).

---

> > ### Comment · Reviewer_ZodG · 2025-08-01
> >
> > Thanks for the detailed discussions on the KS bound. My score remains unchanged.

---

### Official Review · Reviewer_bmYk · 2025-06-23

**Clarity:** 3
**Significance:** 3
**Originality:** 3
**Rating:** 4
**Confidence:** 2

**Summary:**

This paper proposes two sharp, closed-form empirical Bernstein inequalities for symmetric random matrices with bounded eigenvalues. The first inequality (Theorem 3.1) applies to the sample mean of independent matrices and uses a paired sample variance estimator. The second (Theorem 4.2 and Corollary 4.3) extends to martingale-dependent sequences.

**Questions:**

1. Could you suggest any potential applications of the matrix EB inequalities? e.g. for bandit problems or PAC-Bayesian bounds. Including such applications might make this paper more suitable for machine learning conferences.

2. Can you prove that the second EB inequality is sharper than the first one for iid matrices, or only empirically so?

3. Is there a techinical difficulty in extending the first EB inequality to odd numbers? What is the difference between paired sample variance and sample variance in order to give a bound?

4. On the techinical side, what is the main difficulty in extending EB inequality to matrices?

**Ethical Concerns:**

["NO or VERY MINOR ethics concerns only"]

**Final Justification:**

I find the technical analysis convincing and have revised my score accordingly. However, the paper's impact is still limited by the absence of examples in machine learning to demonstrate the importance of the proposed inequalities.

**Limitations:**

yes

**Quality:**

3

**Strengths And Weaknesses:**

The paper is technically sound, and proves novel matrix Empirical Bernstein inequalities that are asymptotically tight. However, the authors only suggest that the matrix EB inequalities can be used to obtain tighter confidence interval, and do not relate it to machine learning problems, so I believe that it is out of the scope of NeuIPS2025 and should be submitted to statistics journals or conferences instead.

---

> ### Author Rebuttal · Authors · 2025-07-30
>
> ## Application
>
> A recurring example of application in the matrix concentration inequality literature is *estimating the covariance matrix* $\Sigma = \mathbb E x_1 x_1^t$ for mean-zero bounded vectors $x_1, \dots, x_n$. See Chapters 1.6.2 to 1.6.4 of J.A. Tropp's monograph _An Introduction to Matrix Concentration Inequalities_ for a classical solution using the (oracle) matrix Bernstein inequality; and Corollary 5 in Howard et al. (2021, cited) for a time-uniform confidence sequence in a similar spirit.
>
>
> In both examples, however, the concentration inequality involves the unknown $\| \Sigma \|$ term itself on the right hand side (i.e. radius) that comes from bounding the variance $\operatorname{Var}( x_1 x_1^t)$ via a basic boundedness argument. Further boundedness argument on $\| \Sigma \|$ is needed if one wants a *closed-form confidence sets for $\Sigma$*, and such an argument essentially reduces the "Bernstein" bound to a Hoeffding bound, and can be arbitrarily loose.
>
> Our EB inequality completely avoids this issue. The radius of the confidence set is a empirical quantity adaptive to $\operatorname{Var}( x_1 x_1^t)$.
>
>
> We have added this example in our paper.
>
> ## Out of scope for NeurIPS
>
> Besides the application of covariance estimation (which is itself a frequently visited topic in NeurIPS, see e.g. "Robust gaussian covariance estimation in nearly-matrix multiplication time" by Li and Ye, "Differentially private covariance estimation" by Amin et al.) mentioned above, the following is an incomplete list of application-driven papers published in NeurIPS that have employed the oracle matrix Bernstein inequality:
>
> - "Provable Non-linear Inductive Matrix Completion" by Zhong et al.
> - "Support Recovery in Sparse PCA with Incomplete Data" by Lee et al.
> - "Distributed estimation of the inverse hessian by determinantal averaging" by Derezinski and Mahoney.
>
> While these papers usually focus on using the matrix Bernstein as an analysis tool, our paper brings uncertainty quantification tools to these applications.
>
>
>
> ## Can you prove that the second EB inequality is sharper than the first one for iid matrices, or only empirically so?
>
>
> On the one hand, both inequalities are "sharp" by our definition in Eqn.(8). It is indeed an empirical observation that the second EB inequality is tighter than the first. There is some understanding of why this is the case in the scalar setting by a very involved analysis of the second-order term (the 1/n term); see the cited arXiv preprint by Shekhar and Ramdas.
>
>
> ## Paired sample variance vs. vanilla sample variance
>
> While it is still possible to obtain *sharp* matrix EB bounds using the vanilla sample variance, existing analysis always suffers from an unsatisfactory lower-ordered term. Such examples include our Appendix B.3, as well as the cited concurrent work by Kroshnin and Suvorikova (a constant 15 exists in their bound).
>
> We acknowledge that the lack of sharp analysis of the vanilla sample variance does not indicate that it is a worse estimator itself. As we mention in the paper, we "leave it to future work whether an improved analysis of the classical matrix sample variance better than (104) can lead to a Maurer-Pontil-style matrix EB inequality that beats Theorem 3.1."
>
> ## Main difficulty in extending EB inequality to matrices
>
>
> For the first EB inequality, the correct variance bound, Eqn.(61), is not straightforward to obtain, as many less involved approaches would end up leading to worse bounds (one of such approaches is recorded in Appendix B.3).
>
> For the second EB inequality, though the scalar version is well-established, it takes much insight to see the scalar self-normalizing technique is applicable to matrices (Lemma 4.1) in a way that is compatible with Lemma 2.2. For example, Howard et al. did not realize such compatibility --- their work contains numerous matrix bounds, even a weaker version of our Lemma 4.1, but not our second matrix EB inequality.

---

> > ### Comment · Reviewer_bmYk · 2025-08-05
> >
> > Thank you for your detailed responses. I find the technical analysis convincing and have revised my score accordingly. However, the paper's impact is still limited by the absence of examples in machine learning to demonstrate the importance of the proposed inequalities. For example, the authors suggest that several papers use the matrix Bernstein to analyze machine learning problems, but it is unclear whether the refined empirical Bernstein inequalities are more beneficial in these settings.

---

### Official Review · Reviewer_FjiK · 2025-06-26

**Clarity:** 3
**Significance:** 2
**Originality:** 2
**Rating:** 4
**Confidence:** 4

**Summary:**

The paper generalizes scalar empirical Bernstein inequalities to the matrix case. Known scalar empirical Bernstein inequalities give a bound on the deviation of the sample mean from the true mean in terms of the empirical variance for i.i.d. samples, or a bound on the deviation of a particular weighted average from the conditional mean in terms of some empirical quantity in the general case. For matrices, a Bernstein type inequality bounds the largest eigenvalue of the centralized average of n i.i.d symmetric matrices in terms of some matrix V that upper bounds the matrices variance w.r.t. to the Loewner partial order. Empirical variants are notoriously harder to attain. The authors formulate a couple of sharp matrix Bernstein empirical inequalities, where sharp means that its deviation terms approache the non-empirical Bernstein inequality deviation term a.s. including constants.

**Questions:**

In eq. (58), where does the factor 2 in the parentheses originate from? This is an average of dxd matrices, so shouldn't be d/alpha instead of 2d/alpha?

**Ethical Concerns:**

["NO or VERY MINOR ethics concerns only"]

**Final Justification:**

The authors convincing rebuttals and discussion with the reviewers improved my opinion of the paper.

**Limitations:**

The authors adequately addressed the limitations and potential negative societal impact of their work.

**Paper Formatting Concerns:**

I did not notice any major formatting issues in this paper.

**Quality:**

3

**Strengths And Weaknesses:**

Strengths: the paper is clear and concise, the discussion on the problem, known results and contributions of the paper are thorough.
Weaknesses: I am not convinced that the results are significant enough, and find the originality of the paper to be lacking. For example, I would like to see more convincing arguments why empirical Bernstein inequalities for matrices are important. In the scalar problem, the empirical inequality allows one to obtain an empirical confidence interval for the true mean given the sample average, when the true variance is unknown. Can something similar be said about the matrix case? As far as originality, it seems to me that the proofs are derivative of previous works, e.g., Theorem 3.1. is obtained by applying the matrix Bennett-Bernstein inequality twice and using the union bound in a non-trivial way.

---

> ### Author Rebuttal · Authors · 2025-07-30
>
> ## Why empirical Bernstein inequalities for matrices are important.
>
>
> In answer to the reviewer's question if the matrix EB inequalities provide, like their scalar siblings
>
> > An empirical confidence interval for the true mean given the sample average, when the true variance is unknown
>
> our Theorem 3.1 does exactly that, where the "confidence interval" is now a confidence sphere in the spectral distance.
>
> Further, the importance of matrix EB inequalities lies in their application. A recurring example of application in the matrix concentration inequality literature is *estimating the covariance matrix* $\Sigma = \mathbb E x_1 x_1^t$ for mean-zero bounded vectors $x_1, \dots, x_n$. See Chapters 1.6.2 to 1.6.4 of J.A. Tropp's monograph _An Introduction to Matrix Concentration Inequalities_ for a classical solution using the (oracle) matrix Bernstein inequality; and Corollary 5 in Howard et al. (2021, cited) for a time-uniform confidence sequence in a similar spirit.
>
>
> In both examples, however, the concentration inequality involves the unknown $\| \Sigma \|$ term itself on the right hand side (i.e. radius) that comes from bounding the variance $\operatorname{Var}( x_1 x_1^t)$ via a basic boundedness argument. Further boundedness argument on $\| \Sigma \|$ is needed if one wants a *closed-form confidence sets for $\Sigma$*, and such an argument essentially reduces the "Bernstein" bound to a Hoeffding bound, and can be arbitrarily loose.
>
> Our EB inequality completely avoids this issue. The radius of the confidence set is a empirical quantity adaptive to $\operatorname{Var}( x_1 x_1^t)$.
>
>
> We have added this example in our paper.
>
> ## The factor 2 in (58)
>
> We need a two-sided bound (as opposed to the one-sided inequality (5)) here. So we split the $\alpha$ two-way, apply (5) twice, and take a union bound.

---

> > ### Comment · Reviewer_FjiK · 2025-08-05
> >
> > Thank you for your response, I find the arguments convincing and have increased my score accordingly.

---

### Official Review · Reviewer_CtZ1 · 2025-07-23

**Clarity:** 3
**Significance:** 3
**Originality:** 3
**Rating:** 4
**Confidence:** 3

**Summary:**

The paper considers the question of proving matrix empirical Bernstein-type concentration inequalities. In its simplest form, the Bernstein inequality is a well-known bound that is used to reason about the empirical mean of independent random variables with common mean and bounded support that satisfy a second moment condition. The second moment condition involves a variance bound $\sigma^2$ which is usually not known in practical settings.
There are a number of generalizations and stronger versions of this result that one can consider. First, it is important to consider ``empirical" versions of this inequality, where the unknown bound $\sigma^2$ is replaced by an estimate for the variance. This has the advantage that the deviation of the empirical mean from the true mean can then be quantified precisely, instead of being expressed as a function of the unknown variance bound. Another generalization involves going beyond independent random variables, and considering various kinds of dependencies. Finally, it is worth considering generalizations from scalars to vectors and matrices.
The present work gives empirical Bernstein-type concentration inequalities for matrices, both for independent random matrices and for random matrices that exhibit a martingale dependence. Both inequalities exhibit sharp constants.

**Questions:**

Can the authors provide a more detailed discussion of the challenges involved in proving the results? Also, can they give more concrete examples of problems where using these concentration bounds would yield significant benefits?

**Ethical Concerns:**

["NO or VERY MINOR ethics concerns only"]

**Final Justification:**

Based on the authors' response, I have greater appreciation for the significance and originality of the work than before. However, that does not change my overall rating, which is still borderline accept.

**Limitations:**

Yes

**Paper Formatting Concerns:**

No concerns.

**Quality:**

3

**Strengths And Weaknesses:**

I think this is an interesting work, but I have some concerns regarding its overall merit. For that reason, I will try to discuss both positive and negative aspects of the work below.

Strengths:
1) The main results are interesting in their own right. Indeed, having empirical versions of standard concentration inequalities generalized to the matrix setting and with sharp constants constitutes an interesting contribution to the literature.
2) The writing is generally accessible.

Weaknesses:
1) It is mentioned that it is generally harder to establish concentration results for random matrices, compared to how hard it is to achieve this for random variables or vectors. However, I felt there was not an adequate discussion of the technical challenges that the authors encountered when attempting to establish the results in this work. Instead, it felt like the results mostly follow some approaches which have appeared in prior work (e.g., Audibert et al. [2009], ], Maurer and Pontil [2009], Tropp [2012], etc).
2) No application of the results is given. The paper makes the case about the importance of the result based on the fact that empirical versions of concentration inequalities are generally useful in practice, and that constants are sharp. However, I feel the authors could have made a stronger case about the importance of the results, by a giving a concrete application.
Overall, I do not consider this to be a bad paper by any means, but I think it might be a bit marginal. I am voting borderline accept for the time being.

J.-Y. Audibert, R. Munos, and C. Szepesvari. Exploration–exploitation tradeoff using variance estimates in multi-armed bandits. Theoretical Computer Science, 410(19):1876–1902, 2009.

A. Maurer and M. Pontil. Empirical Bernstein bounds and sample variance penalization. In 22nd Annual Conference on Learning Theory, 2009.

J. A. Tropp. User-friendly tail bounds for sums of random matrices. Foundations of Computational Mathematics, 12:389–434, 2012.

---

> ### Author Rebuttal · Authors · 2025-07-30
>
> ## The challenges involved in proving the results
>
> We agree with the reviewer that the *first* EB inequality echos much of the prior effort made by Audibert, Maurer and Pontil, Tropp etc. Still, we argue that the correct variance bound, Eqn.(61), is not straightforward to obtain, as many naive approaches would end up leading to worse bounds (one of such approaches is recorded in Appendix B.3).
>
> Our *second* EB inequality is very non-trivial. Though the scalar version is well-established, it takes much insight to see the scalar self-normalizing technique is applicable to matrices (Lemma 4.1) in a way that is compatible with Lemma 2.2. For example, Howard et al. did not realize such compatibility --- their work contains numerous matrix bounds, even a weaker version of our Lemma 4.1, but not our second matrix EB inequality.
>
>
> ## Concrete examples
>
> A recurring example of application in the matrix concentration inequality literature is *estimating the covariance matrix* $\Sigma = \mathbb E x_1 x_1^t$ for mean-zero bounded vectors $x_1, \dots, x_n$. See Chapters 1.6.2 to 1.6.4 of J.A. Tropp's monograph _An Introduction to Matrix Concentration Inequalities_ for a classical solution using the (oracle) matrix Bernstein inequality; and Corollary 5 in Howard et al. (2021, cited) for a time-uniform confidence sequence in a similar spirit.
>
>
> In both examples, however, the concentration inequality involves the unknown $\| \Sigma \|$ term itself on the right hand side (i.e. radius) that comes from bounding the variance $\operatorname{Var}( x_1 x_1^t)$ via a basic boundedness argument. Further boundedness argument on $\| \Sigma \|$ is needed if one wants a *closed-form confidence sets for $\Sigma$*, and such an argument essentially reduces the "Bernstein" bound to a Hoeffding bound, and can be arbitrarily loose.
>
> Our EB inequality completely avoids this issue. The radius of the confidence set is a empirical quantity adaptive to $\operatorname{Var}( x_1 x_1^t)$.
>
>
> We have added this example in our paper.

---

### Decision · Program_Chairs · 2025-09-17

**Decision:**

Accept (poster)

**Comment:**

This paper presents two empirical Bernstein-type inequalities for symmetric random matrices. The bounds are non-asymptotic, do not require prior knowledge of the variance, and are sharp, including the constants.

Four reviews were collected. All reviewers highlighted the importance and significance of the results, particularly the technical contributions. The AC concurs with this assessment and regards the paper as a significant breakthrough on an old yet important problem in statistics. The sharpness of the derived bounds, both for the sample mean of independent i.i.d. matrices and for their martingale-dependent variants, is especially noteworthy.

Some reviewers expressed concerns about the applicability of the results, but the authors provided satisfactory responses to address these points.